# Synthetic urine oversimplification results in misleading membrane fouling mechanisms in bipolar membrane electrodialysis

Hao-Ran Yang[1,2], Shu-Jie Hu[1,2], Meng-Yue Zhang[1,2], Di Wu[1,2], Lei Zheng[1,2], Qianhong She[3,4], Zhi-Hua Yuan[5], Lin-Jiang Zhong[1,2], Xuan Zhao[6], Ying Chen[6], Hong Liu[1,2], Lin-Ji Xu[7] ✉ & Yuan Liu[1] ✉

Lab-scale wastewater treatment studies, including urine recovery, often rely on oversimplified synthetic wastewater, thereby compromising the reliability of results and data. Here, we systematically evaluate how using full-component versus simplified synthetic urine formulations affects the performance and engineering-economic assessment of bipolar membrane electrodialysis. Our findings reveal that simplification fundamentally alters fouling mechanisms. While urea alone causes significant damage to anion exchange membranes through hydrogen bonding aggregation, natural organic co-components in real urine mitigate fouling via inhibitory interactions—a mechanism confirmed by molecular dynamics simulations and experimental characterization. After seven batches, the full-component formulations showed 29.3% less performance decay and 10–14% higher urea recovery than the simplified formulation. Moreover, the complete removal of organic components disrupts ion-organic co-aggregation pathways on cation exchange membranes, shifting fouling toward mineral crystallization on bipolar membranes. Simplification also distorts the engineering-economic assessment, overestimating cleaning costs by 15.9% and underestimating membrane lifespan by 12.5%. These findings necessitate identifying key wastewater constituents and ensuring experimental integrity to bridge lab-industry gaps, advocating designs comprehensively addressing multi-component interactions.

Advances in sustainable water treatment technologies are pivotal in addressing global resource and environmental challenges[1–3]. However, a significant gap remains between fundamental research and engineering practice, particularly in wastewater treatment, where lab-derived findings often fail to translate into reliable real-world applications. Current experimental frameworks commonly rely on simplified synthetic formulations to characterize complex wastewater systems—a pragmatic approach driven by cost efficiency and reproducibility[4,5]. While this allows controlled investigations, it inherently disrupts the interconnected mechanisms of multidimensional

[1]State Key Laboratory of Lake and Watershed Science for Water Security, Chongqing Institute of Green and Intelligent Technology, Chinese Academy of Sciences, Chongqing, China. [2]Chongqing School, University of Chinese Academy of Sciences, Chongqing, China. [3]School of Civil and Environmental Engineering, Nanyang Technological University, 50 Nanyang Avenue, Singapore, Singapore. [4]Singapore Membrane Technology Centre, Nanyang Environment and Water Research Institute, Nanyang Technological University, 1 Cleantech Loop, Clean Tech One, #06-08, Singapore, Singapore. [5]CAS Key Laboratory of Urban Pollutant Conversion, Institute of Urban Environment, Chinese Academy of Sciences, Xiamen, China. [6]School of Environmental Science and Engineering, Southwest Jiaotong University, Chengdu, China. [7]College of Environment and Ecology, Chongqing University, Chongqing, China. ✉e-mail: lin_ji_good@126.com; liuyuan@cigit.ac.cn

interfacial processes (physical, chemical, and biological) that characterize real-world systems. For instance, neglecting trace organics, colloidal particles, and dynamic ionic compositions in synthetic formulations distorts degradation kinetics, misestimates treatment performance, and disrupts microbial community succession[6–8]. Such issues undermine the reliability of performance predictions for industrial-scale systems. Consequently, applying simplified experimental formulations to the inherent complexity of actual wastewater systems poses significant challenges.

This issue is starkly exemplified in the field of urine resource recovery[5,9,10]. Systematic reviews indicate that fewer than 20% of studies employ real human urine, with most relying on synthetic formulations deficient in critical organic metabolites[9,11]. Key omissions (e.g., creatinine and uric acid) disrupt urea hydrolysis, membrane stability, and microbial synergy[12], yet overly idealized synthetic formulations persist, creating an artificial paradigm that misguides decentralized sanitation technology development. In recent years, bipolar membrane electrodialysis (BMED) has demonstrated significant potential in resource recovery by utilizing the hydrolysis properties of bipolar membrane (BPM) under an electric field, enabling the recovery of nutrients from urine in the form of acids, bases, and fertilizers[13–17]. However, these studies predominantly rely on oversimplified synthetic formulations, limiting its guidance for industrial-scale applications.

An illustrative example of this overestimation lies in BMED systems tested with synthetic urine (e.g., $1.75\,g\,L^{-1}$ total nitrogen), which reported high nitrogen removal efficiencies (80–89%) under a current density of $150\,A\,m^{-2}$. These results were overestimated owing to neglecting key organic constituents typically found in actual wastewater systems[18]. The transition to actual urine ($2.0\,g\,L^{-1}$ $Na^+$, $4.9\,g\,L^{-1}$ $NH_4^+$, and $15\,g\,L^{-1}$ $CO_3^{2-}$) revealed the evident limitations of simplified compositions: competitive cation migration ($Na^+$ vs. $NH_4^+$) reduced nitrogen mass transfer efficiency by over 50% due to preferential occupation of cation-exchange membrane channels, while the dynamic carbonate-phosphate buffering system shifted acid chamber pH from synthetic formulation values (pH 1.07) to near-neutral conditions (pH 7.50) through $CO_2$ back-diffusion and phosphate protonation[11,17]. These phenomena are absent in synthetic formulations. Further compounding this issue is the neglect of organic-inorganic interactions. Current studies often isolate inorganic ions (e.g., $3772\,mg\,L^{-1}$ $NH_4^+$ and $3080\,mg\,L^{-1}$ $Cl^-$) or single organic species (e.g., $15\,g\,L^{-1}$ urea), while actual urine comprises a complex matrix of metabolites that synergistically regulate interfacial processes in BMED. Previous research reported that in a bioelectrically enhanced BMED system (BBGS), creatinine acted as an electron shuttle, enhancing microbial urease activity and facilitating the complete hydrolysis of urea ($1717\,mg\,L^{-1}$) to ammonium. Simultaneously, histidine stabilized the pH gradient across BPM via its imidazole group, thereby increasing ammonia recovery to 90.39%[16]. These mechanisms, which rely on the coexistence of organic and inorganic constituents, are absent in the simplified formulations commonly used in studies that neglect cross-component synergy. Such persistent oversimplification distorts the identification of fouling drivers, biasing predictive models toward idealized interactions. Consequently, it creates a fundamental knowledge gap: How do complex component synergies—routinely ignored in oversimplified synthetic formulations—impact treatment performance and obscure experimental interpretability of performance-limiting mechanisms in BMED systems? Critically, the prevailing reliance on oversimplified formulations (e.g., containing only urea) creates a systematic blind spot[5,10,16,17]. This paradigm is fundamentally limited, as it inherently fails to capture the full spectrum of intermolecular interactions—particularly the overlooked antagonistic interactions that, counterintuitively, mitigate rather than exacerbate membrane fouling. This oversight not only obscures key fouling

mechanisms but also risks a complete misdiagnosis of fouling risks in real-world systems.

To address this knowledge gap, this study systematically investigates how compositional fidelity in synthetic urine formulations—ranging from basic salt systems to complex mixtures with organic metabolites—impacts the performance of BMED system. By constructing synthetic urine formulations with graded complexity and integrating system stability testing and engineering-economic assessment, we quantitatively dissect the effects of formulation variations on the efficiency of nutrient recovery. Through experimental characterization and molecular dynamics simulations, this study clarifies the intrinsic correlations between compositional fidelity, nutrient recovery efficiency, and operational stability. This study advocates for a paradigm in water research, where compositional fidelity is a cornerstone of technological validation, addressing critical oversimplifications in environmental engineering water treatment and ensuring robust scalability assessments for next-generation wastewater systems.

## Results
### Urine complexity impacts BMED performance

As demonstrated in Fig. 1a, applied electric fields drive selective ion migration: $H_nPO_4^{(3-n)-}$, $NH_4^+$ and $K^+$ traverse ion-exchange membranes (IEMs) into the acid and base chambers, respectively, where they combine with $H^+/OH^-$ generated by the BPM to produce acid and base. Under alkaline conditions, $NH_4^+$ is converted into gaseous $NH_3$. The $NH_3$ then diffuses through the hollow fiber membrane (HFM) and is absorbed by circulating sulfuric acid in the absorption chamber, forming $(NH_4)_2SO_4$. Meanwhile, the untransferred $H_nPO_4^{(3-n)-}$ and electrically neutral urea remain confined to the feed chamber, where they can be further utilized as raw materials for liquid fertilizer production.

To bridge the gap between synthetic and real wastewater, the formulations were designed based on comprehensive analysis of urban urine composition (Supplementary Table 1-2). Group A represents the conventional simplified formulation[9,17], containing only dominant inorganic salts (e.g., NaCl, KCl) and urea. Group B expands this by incorporating core small organic metabolites—creatinine and uric acid—which are not only universally present in actual urine but are also consistently employed in rigorous studies to enhance the representativeness of synthetic urine formulations[5,16,17]. Group C further introduces macromolecular organics, using Bovine Serum Albumin (BSA) as a well-established model protein. This choice is justified by its high structural and functional similarity to human serum albumin, a key proteinaceous component in urine, allowing for a controlled investigation of macromolecular fouling mechanisms relevant to real systems[19–21]. This hierarchical design covers key categories of actual urine components—from ions to small organics and macromolecules—enabling a comprehensive analysis of their synergistic effects. Under a constant voltage of 15 V and a flow rate of $0.4\,L\,min^{-1}$, desalination efficiencies of 85.9%, 95.4%, and 95.4% were obtained from the first batch of 2-hour experiments for Groups A, B, and C, respectively (Fig. 1b). However, performance progressively degraded in subsequent batches, with efficiencies dropping to 55.7%, 67.4%, and 69.0% by the seventh batch. Notably, the ratio of performance decay was inversely related to compositional complexity (A < B < C), with decay ratios of 54.2%, 41.4%, and 38.3%, respectively (Fig. 1c). Despite the differential performance decay, the core functionality of the BMED process remained relatively stable, as reflected in the consistent pH evolution of the key chambers (Supplementary Fig. 1). Both the acid and base chambers established their respective strongly acidic (pH <2.5) and alkaline (pH > 10.5) environments within the first 30 minutes of operation and maintained them consistently throughout the experiments, providing a stable driving force for nutrient recovery and

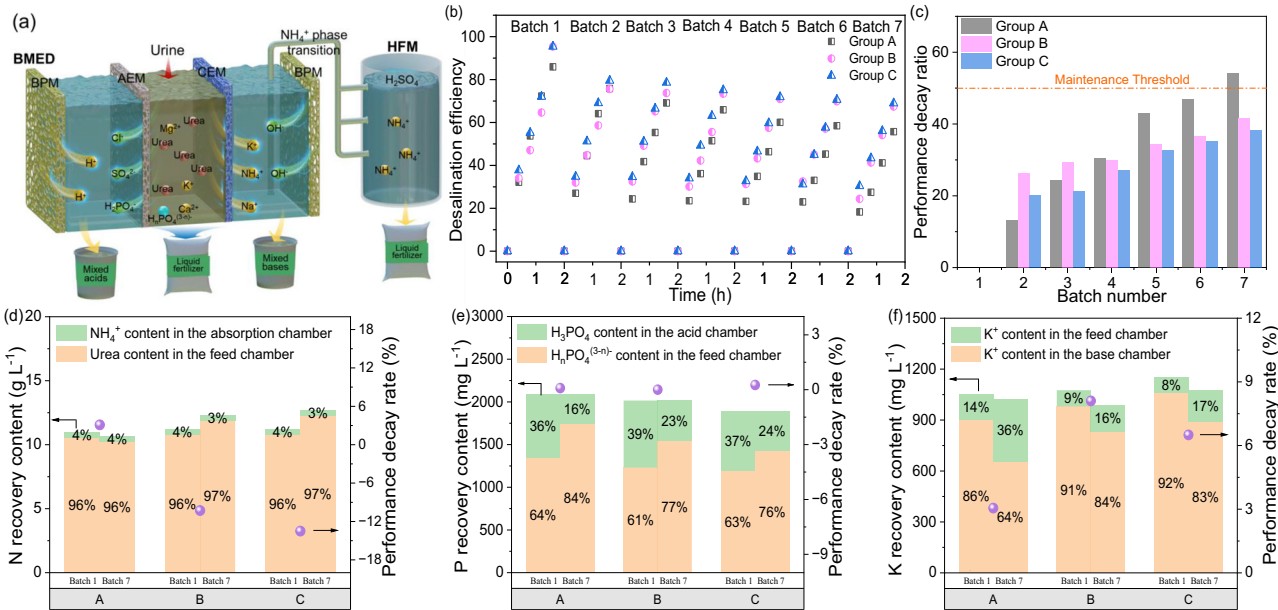

**Fig. 1 | Impact of synthetic urine complexity on the BMED system performance over seven batches. a** Schematic of the nutrient recovery mechanism via the BMED system. **b** Variation in desalination efficiency, **c** System performance decay ratio, **d** Nitrogen recovery content and its decay rate, **e** Phosphorus recovery content and its decay rate, **f** Potassium recovery content and its decay rate. BPM: bipolar membrane; AEM: anion-exchange membrane; CEM: cation-exchange membrane; HFM: hollow fiber membrane; Feed chamber: urine inflow; Acid chamber: $H^+$ from BPM hydrolysis; Base chamber: $OH^-$ from BPM hydrolysis; Absorption chamber: The gas-liquid mass transfer interface constructed by HFM achieves selective $NH_3(g)$ capture and chemical fixation using preloaded sulfuric acid as the absorption medium. Data points represent measurements from a single, continuous operational sequence for each group.

ammonia stripping. The feed chamber pH remained stable at approximately 3, a consequence of inherent proton leakage in the BMED process[22]. This phenomenon, however, conferred a critical advantage by serendipitously suppressing urea hydrolysis, thereby obviating the need for exogenous acid addition and significantly facilitating the storage and transportation of urine resources[23]. While classical membrane fouling theory suggests that increased compositional complexity exacerbates synergistic fouling (e.g., organic-inorganic co-deposition[24,25]), our results revealed a counterintuitive trend: higher formulation complexity correlated with slower performance decay. Specifically, Group A (simplified formulation) exhibited a > 50% performance decay by the seventh batch, exceeding typical maintenance thresholds for acceptable decay, whereas Groups B and C showed more gradual performance degradation (Fig. 1c).

Subsequently, this study further examined the cascade effects of compositional complexity on resource recovery efficiency in the BMED system (Fig. 1d-f). As operational batches increased, the system exhibited performance changes, with desalination efficiency declining by 26.4–30.2%, resulting in a corresponding reduction in overall nutrient recovery efficiency. As shown in Fig. 1d, $NH_4^+$ recovery remained largely unaffected by performance decay due to its stable transmembrane migration, a finding consistent with results from previous studies[17,26]. In contrast, urea recovery efficiency exhibited significant variability across experimental groups: Group A (urea-only) showed a 3.2% decline, whereas Groups B and C, supplemented with creatinine, uric acid and BSA, demonstrated 10–14% improvements in recovery efficiency. These findings suggested that the addition of small and large molecules in Groups B and C contributed to the stabilization of system performance, potentially by reducing urea migration through specific interactions that limit its transmembrane passage. Further analysis revealed that the transmembrane migration ratios of phosphorus (P) and potassium (K) in the seventh batch increased from 16% and 64% in Group A to 23% and 84% in Group B, and further to 24% and 83% in Group C, respectively (Fig. 1e,f). These results suggested that the complex interactions within Groups B and C stabilized BMED

performance, thereby facilitating electromigration-driven transmembrane transport of ions such as $H_nPO_4^{(3-n)-}$ and $K^+$. This enhancement in ion transport directly facilitated the generation of high-value acids and bases in the BMED system, not only improving urinary nutrient recovery efficiency but also enhancing system economic viability. Pressure-driven technologies such as nanofiltration and reverse osmosis exhibit heightened sensitivity to compositional complexity, with studies[27–29] demonstrating severe flux declines (e.g., 65% reduction with 1 mM calcium ion[30], synergistic organic-inorganic fouling[27]). Such discrepancies highlighted a critical knowledge gap: over-simplified urine compositions disproportionately amplify single-component fouling risks (e.g., exaggerated treatment performance decline in Group A), while failing to capture emergent stabilizing interactions in multi-component systems. These findings suggested that compositional complexity likely stabilizes BMED performance through interactions among urinary components—a mechanism often overlooked in simplified formulations—that may extend to other complex aqueous systems rich in organics and inorganics.

While traditional membrane systems exhibited performance losses due to actual wastewater complexities, our research revealed an inverse relationship, where increased complexity enhances operational stability. Both phenomena converged on a central idea: over-simplified compositions can lead to misdiagnosis of fouling mechanisms. These insights necessitated integrative approaches to wastewater treatment optimization, where component interaction mapping supersedes single-factor fouling assessments.

## Organic interactions reduce anion-exchange membrane fouling

Scanning electron microscopy coupled with energy-dispersive X-ray spectroscopy (SEM-EDS) investigates the spatial distribution and semi-quantitative elemental composition of the fouling layer of the anion-exchange membrane (AEM) exposed to various synthetic urine formulations. Obviously, the surface deposits of the BMED-treated groups were predominantly composed of C, N, and O (Fig. 2a,b), indicating preferential adsorption of nitrogen-rich organic

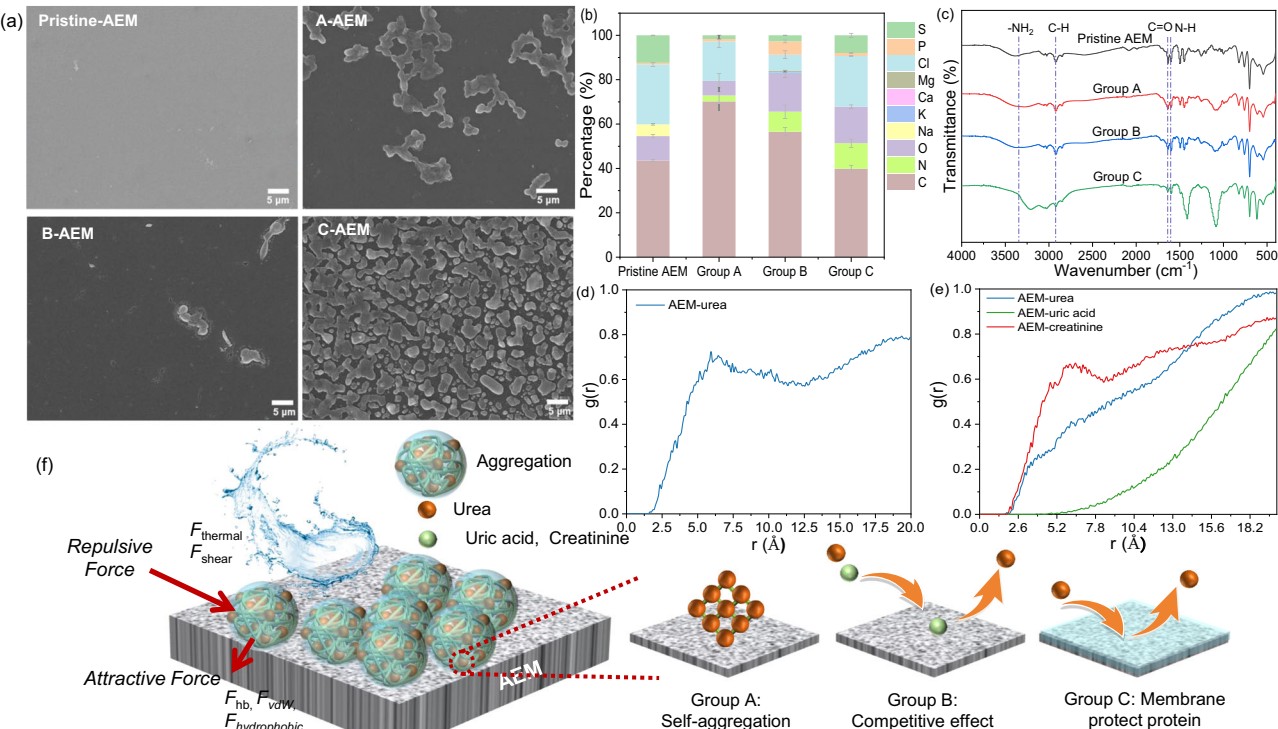

**Fig. 2 | AEM fouling in the BMED system during urine nutrient recovery from different groups over seven batches. a** SEM morphological analysis (In Group A (urea-only), spherical primary urea aggregates constitute the fouling layer. In Group B (with creatinine/uric acid), the density of urea aggregates is significantly reduced due to competitive inhibition, resulting in a cleaner membrane surface. In Group C (with BSA), a uniform proteinaceous layer predominates, demonstrating steric hindrance.); **b** EDS quantitative detection of fouling components. Error bars represent the standard deviation (s.d.) of $n = 2$ measurements taken at different locations on the membrane surface to account for spatial heterogeneity; **c** Fourier transform infrared spectroscopy (FTIR) characterization; Molecular dynamics (MD) simulation: **d** Radial distribution function (RDF) analysis of interactions between AEM and urea; **e** RDF analyses of interactions between urea, uric acid, creatinine and AEM; **f** Fouling aggregation mechanisms, fouling mechanisms and antagonistic mechanisms: Urea self-aggregates at the AEM interface, with hydrogen bonding ($F_{hb}$), van der Waals forces ($F_{vdW}$), and hydrophobic interactions ($F_{hydrophobic}$) being the primary forces maintaining the aggregates, while thermal ($F_{thermal}$) and shear energies ($F_{shear}$) are the energies that need to be overcome; Creatinine and uric acid reduce urea self-aggregation by competing for urea hydrogen bonding sites, whereas BSA prevents urea from contacting the membrane through steric hindrance effects.

compounds onto the AEM surface. A comparative analysis further demonstrated that Group A (urea-only) exhibited severe membrane fouling, characterized by a significantly higher density of particulate deposits on the AEM surface compared to other groups. This phenomenon was directly associated with the observed over 50% performance decay in Group A after seven batches of operation (Fig. 1c). In contrast, Group B, which introduced creatinine and uric acid in addition to urea, exhibited a marked reduction in surface deposit density on the AEM compared to Group A. This suggested that the coexistence of small-molecule organics suppressed pollutant accumulation, thereby maintaining the treatment stability of the BMED system. When BSA was added to form Group C, the distribution of sulfur elements (Fig. 2b, Supplementary Fig. 2) corresponded to cysteine residues in the BSA molecule, indicating that proteins participated in the fouling process through interfacial adsorption. Notably, in Group A, spherical aggregates with a diameter of approximately 1.4 μm were observed (Fig. 2a), three orders of magnitude larger than the theoretical diameter of urea molecules (~0.4 nm)[31,32]. Since Group A contained only urea as the sole organic component, these aggregates were likely urea self-aggregates. Previous studies confirmed through molecular dynamics simulations that urea forms self-aggregates in aqueous solution via hydrogen bond weakening and hydrophobic interactions, but such aggregation phenomena only occur at higher concentrations (>1 M)[32,33]. However, this study demonstrated that membrane interfaces could induce urea molecules to overcome the random aggregation patterns of bulk solutions, forming micrometer-sized structures

through interface-induced nucleation. The phenomenon suggested that the AEM surface provided nucleation sites, enhancing intermolecular interactions between urea molecules and the AEM surface. X-ray diffraction detected characteristic crystal plane signals, and EDS showed N enrichment in the aggregated regions, collectively validating the interface-induced urea self-aggregation process (Fig. 2b, Supplementary Fig. 3).

To elucidate the molecular mechanism of urea self-aggregation at membrane interfaces, this study employed EDS, Fourier transform infrared spectroscopy (FTIR), X-ray photoelectron spectroscopy (XPS) and Molecular dynamics (MD) simulations (Fig. 2b-e and Supplementary Fig. 4-5). FTIR analysis revealed significant spectral shifts, indicating critical hydrogen-bonding interactions: the N-H stretching vibration exhibited a marked redshift ($\Delta v \approx 100$ cm$^{-1}$) from its characteristic 3500 cm$^{-1}$ position, demonstrating hydrogen-bond donation from -NH$_2$ groups of urea to acceptor sites on the AEM surface. Concurrently, reduced intensities of C = O (1640 cm$^{-1}$) and N-H (1600 cm$^{-1}$) bands in urea-fouled AEM (Group A) compared to pristine AEM confirmed that carbonyl and amine groups governed the hydrogen-bonding interactions, facilitating the assembly of cross-linked molecular networks (Fig. 2c). These findings were consistent with previous studies[34,35], providing molecular-level evidence for interface-induced urea self-aggregates. XPS analysis complemented the findings with intensity increases at specific binding energies: ~288.1 eV (C = O) and ~286.2 eV (C-N) collectively confirming cross-linked structure formation via hydrogen bonding in urea molecules (Supplementary Fig. 4).

MD simulations served as a visual complement to the mechanisms revealed by experimental characterizations, with snapshot images clearly depicting the aggregation of urea molecules on the AEM surface (Supplementary Fig. 5). Radial distribution function (RDF) analysis revealed a prominent peak at the interface of the AEM, indicating a preferential accumulation of urea molecules at this distance scale on the AEM surface, thereby highlighting a significant interaction between urea and the AEM interface (Fig. 2d). This conclusion was consistent with experimental characterization results. Furthermore, the peak observed at 5.9 Å in the RDF significantly exceeded the typical hydrogen bond length range (2.5–3.5 Å), suggesting that, in addition to hydrogen bonding, other interactions, such as van der Waals forces, may have also played a role in facilitating the aggregation of urea on the membrane surface. Thus, the urea -$NH_2$ groups preferentially interacted strongly with the AEM interface via hydrogen bonding, driving a specific molecular orientation at the interface: the polar -$NH_2$ groups anchored onto the AEM surface, with the C = O groups oriented outward. These outward-facing C = O groups then served as binding sites for the -$NH_2$ groups of adjacent urea molecules, forming spherical aggregates through hydrogen bonding. The reason for the lower N content in the surface of AEM in Group A compared to Groups B and C can also be attributed to this mechanism, as observed in the EDS (Fig. 2b).

Building on these results, this study further examined the multi-scale interplay governing the stability of urea aggregation on the membrane under BMED treatment conditions, including acid-enhanced hydrogen bonding, van der Waals forces, hydrophobic effects, shear dissipation, and thermal perturbations (details in Supplementary Method 1).Theoretical simulations demonstrated that although the total hydrogen-bonding energy ($1.57 \times 10^{-9}$ J) was insufficient to counteract thermal and shear disturbances ($1.77 \times 10^{-9}$ J), its strong directional preference and high binding energy drove the formation of a cross-linked hydrogen-bonded network within urea aggregates. This was supported by MD simulations and experimental characterizations (Fig. 2c-f, Supplementary Fig. 5). The network achieved dynamic stability by reducing Helmholtz free energy, while protonation in acidic microenvironments further enhanced binding energy[36,37]. Notably, van der Waals interactions and hydrophobic effects ($3.28 \times 10^{-10}$ J) synergistically reinforced network integrity, enabling urea aggregates to maintain a characteristic size ( > 1.4 μm). Consequently, the stabilization mechanism consisted of two complementary components: hydrogen-bond networks provided molecular recognition and orientation constraints, while long-range weak interactions formed an energy buffer system. Their multiscale coupling established a dynamic equilibrium resistant to shear and thermal fluctuations.

The introduction of creatinine and uric acid (Group B) reduced membrane fouling compared to Group A (Fig. 2a). SEM-EDS analyses showed suppressed urea self-aggregation in Group B, consistent with the inverse relationship between system component complexity and performance decay observed in the BMED system (Fig. 1b-c). Furthermore, in Group B, the RDF peak between urea and the AEM surface vanished, being replaced by a peak for creatinine and AEM (Fig. 2e), indicating that the addition of creatinine weakens the interfacial interaction between urea and AEM. This observation suggested that hydrogen bond competition involving creatinine disrupted the hydrogen bonding between the urea -$NH_2$ groups and the membrane surface, thereby inhibiting urea self-aggregation. Additional evidence for this mechanism came from FTIR spectral shifts—the narrowing of the $NH_2$ peak and a blueshift—indicating competitive hydrogen bonding via preferential adsorption of other nitrogen-containing groups (such as =NH, -NH-) from creatinine and uric acid (Fig. 2c). XPS further supported this mechanism: attenuated C-N ( ~ 286.2 eV) signals and the varying intensity of the -$NH_2$ ( ~ 401.6 eV) peak,

reflecting the competitive occupation of adsorption sites, confirmed the disruption of urea hydrogen-bond networks (Supplementary Fig. 4). In the treatment of Group C AEM, SEM-EDS confirmed the uniform coverage of the membrane surface by a BSA layer, with sulfur distribution matching BSA composition (Fig. 2a-b, Supplementary Fig. 2). FTIR spectroscopy revealed a redshift of the -$NH_2$ stretching vibration to 3210 cm$^{-1}$ upon BSA adsorption, indicative of hydrogen bonding (N-H···O) restricting vibrational modes of -$NH_2$. This observation agreed with previous studies[38,39]. Concurrently, intensified absorption in the 1000–1300 cm$^{-1}$ region was observed, suggesting stabilization of surface groups through hydrogen-bonded BSA overlayers. XPS analysis demonstrated a downward shift in the C = O binding energy from 531.78 eV to 531.48 eV (Supplementary Fig. 4), implying enhanced electron density around oxygen atoms. These results were consistent with the anti-fouling mechanism of membrane protein coatings[40–42]: (1) Charge shielding through hydrogen bonding between BSA amino groups and membrane oxygen-containing groups (C = O), which neutralized charge activity; and (2) Steric exclusion from the dense BSA layer, physically blocking pollutant access. Although previous studies[43,44] have suggested that BSA fouling may negatively affect ion transport, our results indicate that, compared to urea self-aggregation fouling, BSA fouling exerts a lesser impact on membrane performance. Besides, BSA-fouled membranes (Group C) exhibited higher hydrophilicity than urea-fouled (Group A) counterparts (Δ surface free energy: +13.18 mJ m$^{-2}$, Supplementary Fig. 6), facilitating enhanced ion transport (Fig. 1b-c). Thus, the BSA-fouling not only enhanced ion transport but also suppressed urea fouling, further stabilizing BMED performance.

In summary, the various interactions occurring at the urea-membrane interface and the resulting self-aggregation phenomenon were the primary factors responsible for the significant performance decay of the BMED system. Simplified formulation (Group A) neglected antagonistic interactions among urinary inherent organic molecules—such as competitive adsorption between urea, creatinine, and uric acid, and steric exclusion by proteins—thereby overestimating the individual fouling risk of urea while masking the anti-fouling mechanisms inherent in complex components (Fig. 2f). This oversight biased lab-scale interpretations by amplifying fouling risks of single compounds and obscuring the synergistic anti-fouling effects in actual urine. These differences underscored the limitations of over-simplified wastewater component models for BMED systems, as their lab-level data failed to guide practical industrial applications.

## Organic-inorganic synergy shifts membrane fouling patterns

During urine treatment via the BMED system, both cation-exchange membrane (CEM) and AEM exhibited substantial urea aggregation, while the underlying mechanisms differed. Similar to AEM, Groups B and C (containing creatinine/uric acid or BSA) exhibited a marked reduction in urea-only (Group A) fouling on the surface of CEM (Fig. 3a), reaffirming the inherent antifouling synergy of natural urine components. However, the CEM surface displayed distinct fouling characteristics: SEM imaging showed that the fouling particles were more compact and larger compared to those on AEM. EDS analysis further indicated a significant enrichment of hardness ions ($Ca^{2+}$, $Mg^{2+}$) in CEM deposits (Fig. 3b), with $Ca^{2+}$ concentrations ranging from 0.5-3 wt% and $Mg^{2+}$ from 1-6 wt% across Groups A, B, and C, providing semi-quantitative evidence that these ions were critical mediators of foulant formation. Prior studies[45,46] proposed that hardness ions act as mineralization cores, attracting organic molecules like urea to establish aggregation layers—a mechanism corroborated by our findings. Additionally, these ions also induced electron cloud redistribution in organic molecules, thereby stabilizing extended aggregation structures[47]. Such ion-bridging effects, which synergize urea self-aggregation and hardness ion co-aggregation, led to the formation of

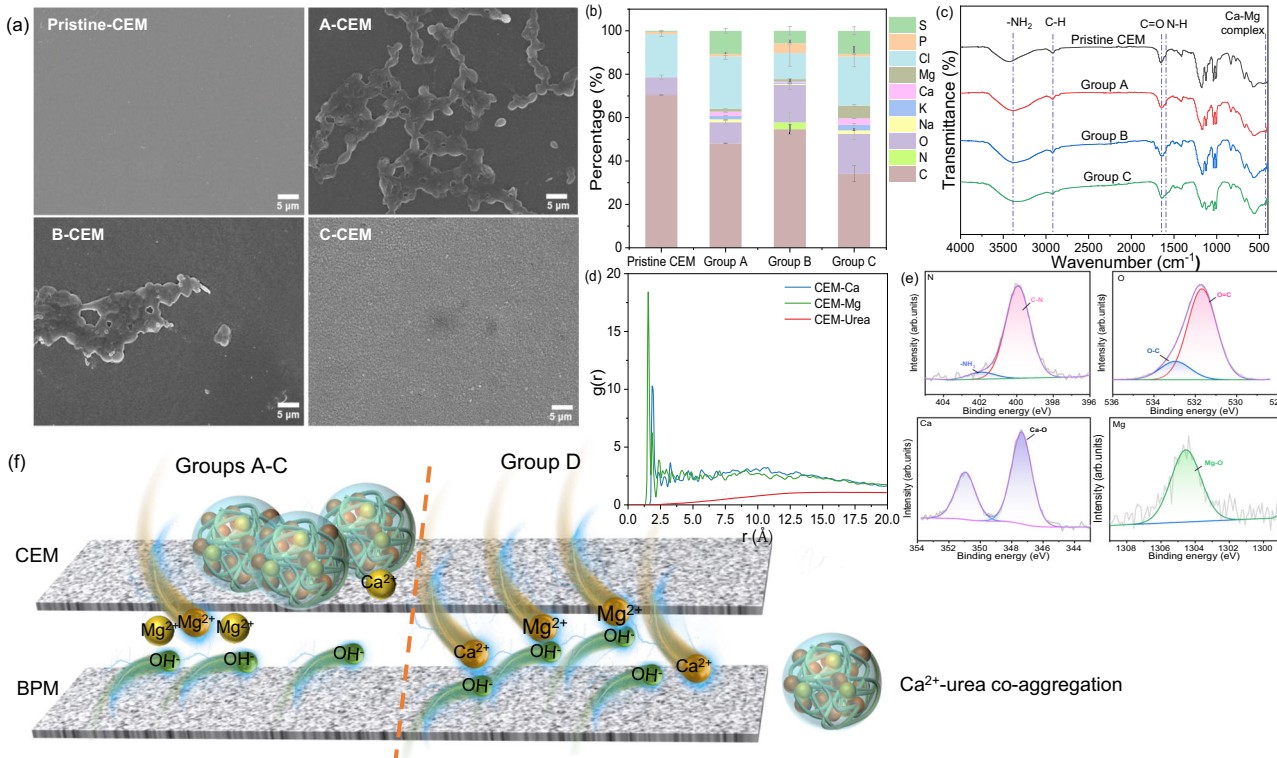

**Fig. 3 | CEM fouling in the BMED system during urine nutrient recovery from different groups over seven batches. a** SEM morphological analysis (Dominant $Ca^{2+}$-$Mg^{2+}$-urea co-deposits in Group A; a heterogeneous composite incorporating creatinine and uric acid in Group B; and foulants superseded by a BSA-$Ca^{2+}$-$Mg^{2+}$ complex in Group C.); **b** EDS quantitative detection of fouling components. Error bars represent the standard deviation (s.d.) of *n* = 2 measurements taken at different locations on the membrane surface to account for spatial heterogeneity; **c** FTIR characterization; **d** RDF analyses of interactions between urea, Ca, Mg and

CEM; **e** XPS analysis of urine fouling in Group A; **f** Scaling site variation and mechanisms: In urea-containing groups (Groups A-C), $Ca^{2+}$ co-aggregates with urea, forming deposits predominantly on the feed side of CEM. In contrast, in the inorganic-only group (Group D), $Ca^{2+}$ rapidly migrates through the CEM and deposits on BPM, triggered by precipitation reactions with $OH^-$ generated by BPM. $Mg^{2+}$ consistently forms scaling on the alkaline side of CEM through interaction with $OH^-$ during its transport across the membrane.

larger, more densely distributed foulant particles on CEM surfaces, ultimately compromising membrane ion-transport efficiency and long-term operational stability.

FTIR and XPS analyses revealed the urea aggregation mechanisms on CEM surfaces and their synergistic interplay with $Ca^{2+}$ and $Mg^{2+}$. FTIR spectroscopy demonstrated that the absorption intensities of C = O ( ~1650 cm$^{-1}$) and -NH$_2$ stretching (~3400 cm$^{-1}$) vibrations on the CEM surface in Groups A-C were readily detectable and more intense than those of the pristine membrane (Fig. 3c), indicating hydrogen bond formation between the carbonyl/amino groups of urea and the oxygen atoms of sulfonate groups (-SO$_3^-$), which are inherent to the CEM. The broadening and intensification of the N-H bond peak (~1600 cm$^{-1}$) further evidenced hydrogen-bond-driven urea adsorption and local aggregation on the CEM. Combined with EDS and XPS, urea-aggregated regions on the CEM surface exhibited spatial co-aggregation of $Ca^{2+}$ and $Mg^{2+}$(Fig. 3b, Fig. 3e and Supplementary Fig. 7). The chemical shift and intensification of the O *1s* C = O peak corroborated carbonyl oxygen coordination with hardness ions, while the Ca *2p* features at ~347.4 eV and ~350.9 eV, indicative of Ca-O coordination with organic ligands, provided direct evidence for metal-ligand bond formation. This coordination likely facilitated the densification of hydrogen-bond networks, as indicated by the shifted C-N/C=O components in the C *1s* spectrum and reconfigured -NH$_2$/C-N signals in the N *1s* spectrum. MD simulations visually demonstrated the co-aggregation of hardness ions and urea molecules on the CEM surface, with RDF showing pronounced interaction maxima for CEM·$Ca^{2+}$ and CEM-$Mg^{2+}$ (Fig. 3d and Supplementary Fig. 8), indicative of exceptional compactness and strong specific coordination[48–51]. The consistent

findings from EDS, FTIR, and XPS collectively suggest that ionic electrostatic interactions and metal-ligand coordination jointly mediated the adsorption of $Ca^{2+}$ and $Mg^{2+}$ on CEM, which in turn promoted urea aggregation via two pathways: (1) direct coordination between metal ions and carbonyl/amino groups of urea, and (2) intermolecular bridging by charged ions to stabilize urea aggregates.

To investigate hardness ion roles, this study designed Group D (inorganic-only urine, Supplementary Table 2) and conducted seven batch continuous experiments. Unexpectedly, BPM exhibited severe scaling in Group D (Supplementary Fig. 9a). This finding stood in stark contrast to the results observed with organic-containing formulations (Groups A-C), where BPM remained largely unscaled (Supplementary Fig. 9b). SEM mapping revealed ion-specific deposition patterns: $Ca^{2+}$ forming calcium hydroxide scale on the BPM anion side, while $Mg^{2+}$ forming magnesium hydroxide on the CEM base chamber side (Supplementary Figs. 9 and 10; as detailed in Supplementary Note 1). This divergence arose from distinct coordination chemistries: $Ca^{2+}$ (ionic radius: 0.99 Å, with lower charge density and higher polarizability[52,53]) preferentially bound urea's groups as nucleation cores in organic-containing formulations (Groups A-C), forming steric aggregates that anchored ions to CEM. The distinct hydration properties of $Ca^{2+}$ and $Mg^{2+}$ further governed their migration paths. $Mg^{2+}$ (ionic radius: 0.71 Å), with its higher charge density, possesses a larger hydration shell and higher hydration energy (-1922 kJ mol$^{-1}$ for $Mg^{2+}$ vs -1577 kJ mol$^{-1}$ for $Ca^{2+}$), which restricted its mobility and enhanced its retention on the CEM[17,54]. This preference, in turn, conferred greater mobility upon the more weakly hydrated $Ca^{2+}$, facilitating its transit across the CEM and subsequent precipitation at the BPM interface. Conventional

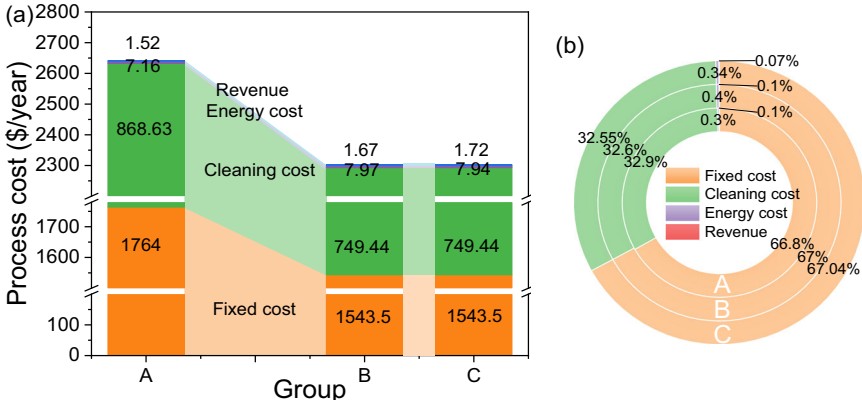

**Fig. 4 | Impact of formulation simplification on engineering-economic assessment. a** Quantitative economic analysis; **b** Proportional cost breakdown.

theories suggest that low fouling on BPM is due to IEMs barrier effects and its self-cleaning via H+/OH- generation[15,55,56]. As a result, in the inorganic-only Group D, mobile $Ca^{2+}$ migrated across the CEM, reacting with OH- generated by the BPM to form scales, which then deposited on its surface. $Mg^{2+}$, however, adsorbed to CEM via electrostatic interactions, regardless of urea presence. These findings indicated that Group D (inorganic-only), compared with Groups A-C (organic-containing), exhibited discrepancies in assessing fouling risk and locations, further revealing the risks of oversimplifying urine components.

Key findings from Group D (inorganic-only) experiments highlighted the essential role of organic-inorganic interactions in governing fouling pathways. Unlike organic-containing Groups A-C, where $Ca^{2+}$ and $Mg^{2+}$ stabilized urea aggregates on CEM via electrostatic bridging and coordination, Group D exhibited severe scaling on BPM, driven by freely migrating $Ca^{2+}$ forming inorganic deposits (Fig. 3f). This underscored that organic (such as urea) acted as both an adsorption matrix and ion anchor: in its absence, hardness ions lost stabilization, leading to unpredictable scale migration (e.g., BPM anion-side $Ca^{2+}$ accumulation vs. CEM-based $Mg^{2+}$ adsorption). Critically, many studies[5,9,10,18] suffered from limitations in experimental design by using inorganic-only urine components similar to those in Group D. This oversimplified experimental design, neglecting organic-inorganic interactions, distorts fouling mechanism interpretation, compromising experimental integrity and yielding unreliable predictions of membrane performance in treating actual wastewater.

### Economic consequences of formulation simplification

The engineering-economic assessment of electrochemical membrane systems is closely tied to their long-term stability[57]. To demonstrate how formulation simplification can systematically bias such assessments, we performed a proof-of-concept cost-revenue analysis building on prior research and based on our lab-scale data (Fig. 4a and Supplementary Table 3)[17,58,59]. This analysis is not intended as a precise industrial forecast, but to quantify the economic risks of relying on oversimplified compositions. Building on previous studies[17,57,60,61], our cost-revenue analysis considered urine compositional complexities. This study provided a comprehensive assessment of energy costs, cleaning costs, fixed costs, and revenue dynamics in the treatment of urine using the BMED system. As shown in Fig. 4b, fixed costs dominate the total expenditure, accounting for approximately 67%, with membrane costs representing the largest share—a finding consistent with prior studies[60,62]. Using mathematical modeling, we predicted cleaning frequencies at the lab-scale (Supplementary Method 2), revealing that cleaning costs constitute approximately 32% of total costs, with modeling approaches diverging between experimental groups. Group A followed an asymptotic decay pattern, while Groups B and C

exhibited multi-stage nonlinear dynamics, requiring fifth-order polynomial models that reflect composition-driven interaction complexities in their systems (Supplementary Fig. 11 and Supplementary Table 4). This increase in cleaning costs was mainly attributed to the amplified fouling effects inherent in small-scale systems[57]. While energy costs and revenue remained relatively modest at this stage, the scalability of the process offers significant economic potential[17]. As production scales up, fixed costs are expected to be amortized over larger outputs, reducing unit costs and enhancing economic viability. Furthermore, the effective recovery of resources and generation of high-value products ensures long-term economic sustainability, a conclusion supported by multiple studies[63,64].

The engineering-economic assessment also revealed a counterintuitive relationship between urine component complexity and operational costs in BMED. As synthetic urine composition transitioned from simplified (Group A) to complex formulations (Group C), membrane cleaning intervals were extended from 12 to 14 hours, reducing annual cleaning batches from 634 to 547 and resulting in a stepwise decrease in total process costs (from $2638.27 per year to $2299.16 per year). Although the complex components (Groups B and C) exhibited higher product yields (Group B: 9.74 kg m-3, Group C: 10.05 kg m-3), representing an 8.8%–12.3% increase compared to Group A (8.95 kg m-3), their annual energy consumption (Group B: 7.97 kWh, Group C: 7.94 kWh) was 10.89%–11.31% higher than that of Group A (7.16 kWh). This was due to the prolonged operation time resulting from reduced cleaning frequency. In contrast, the higher product revenues of Group B ($1.67 year-1) and Group C ($1.72 year-1) compared to Group A ($1.52 year-1) underscored the economic advantages of more complex systems. Notably, the higher cleaning frequency of Group A resulted in higher cleaning costs of 868.63 $ year-1, significantly surpassing those of Groups B and C (749.44 $ year-1). Furthermore, the higher frequency of cleaning triggered a cascade effect, leading to a marked reduction in the lifespan of the membranes, membrane stacks, and peripheral equipment, thus raising fixed costs (from $1543.5 to $1764 per year). This explains why, in the BMED system, the proportions of components in engineering-economic assessment remain consistent despite varying levels of membrane fouling (Fig. 4b).

The engineering-economic assessment further underscored that extrapolating laboratory data derived from simplified formulations to real-world engineering applications could lead to severe misjudgments of long-term operational costs and profits. For instance, cost analyses based on Group A data underestimated the energy requirements in actual urine (where Group A's consumption was 9.82% lower than Group C's), which could potentially distort the predicted return on investment during the pilot phase. Therefore, molecular interactions among urine components not only directly impacted the

operational performance of BMED by governing membrane fouling mechanisms but also indirectly influenced the economic projections of the entire system through system stability.

## Discussion

This study highlights a critical yet overlooked limitation in membrane-based source-separated urine treatment research, demonstrating using the case of BMED systems that oversimplified synthetic compositions fundamentally distort membrane fouling mechanisms and compromise the translational validity of laboratory findings. By systematically constructing the complexity gradient of urine, this study reveals that oversimplified urine formulations exaggerate the fouling risk of an individual compound (urea) and obscure the synergistic anti-fouling effects present in actual wastewater, leading to biased interpretations at the laboratory scale. Specifically, urea self-aggregates at the surface of the IEMs through hydrogen bonding and other interactions, as demonstrated by experimental characterization and molecular dynamics simulations. However, trace organic compounds inherent in urine (such as creatinine, uric acid) and proteins stabilize BMED performance by inhibiting urea-driven self-aggregation through competitive hydrogen bonding and steric hindrance. These inhibitory interactions reduce BMED decay rate by 29.3% at seven batches in multi-component formulations compared to simplified formulations, challenging the conventional assumption that compositional complexity inherently exacerbates fouling. In contrast to the prevailing focus on synergistic fouling, this work establishes that antagonistic organic interactions decisively stabilize BMED performance. This finding challenges the paradigm that wastewater complexity inherently compromises membrane operation and provides a mechanistic foundation for predicting fouling in real-world systems. Furthermore, hardness ions ($Ca^{2+}$ and $Mg^{2+}$) exhibit dual roles: stabilizing urea aggregates through organic-inorganic electrostatic bridging and coordination in organic-containing formulations on the CEM, while triggering mineral scaling in inorganic-only formulations on BPM. Notably, misjudgements in the fouling mechanisms directly lead to performance evaluation biases in the BMED system, subsequently affecting the pre-assessment of its economic viability and stability in industrial applications. These mechanistic differences further demonstrate that simplified formulations, by neglecting component interactions, result in an overestimation of cleaning costs by 15.9% and an underestimation of membrane lifespan by 12.5% in engineering-economic assessment.

When scaled to industrial levels, these systematic errors would cascade into substantial financial miscalculations for full-scale plants, distorting projections of operating expenses, misguiding capital investment decisions, and ultimately misrepresenting the economic competitiveness of the BMED technology. It is crucial to note that this cost analysis, while revealing a systematic bias, is derived from lab-scale operations. To assess the robustness of our finding—that simplified formulations overestimate costs—we first dissected the cost structure. As synthesized in Supplementary Fig. 12, the complex formulation (Group C) achieves a lower total cost primarily by reducing the costs associated with membrane replacement and cleaning (both fixed and operational costs). The economic advantage of Group C lies in its more stable operation, which persists even though it incurs marginally higher energy costs due to longer runtime. We then performed a single-parameter sensitivity analysis on key parameters (e.g., membrane lifespan, energy cost) using industrially-informed variation ranges (Supplementary Table 5)[65]. The result confirms that the economic superiority of Group C remains robust across all tested scenarios. While a full industrial assessment must consider additional scale-dependent factors such as electrode degradation and flow distribution challenges, our lab-scale model unequivocally demonstrates that compositional simplification is a fundamental source of economic misjudgment by

distorting the key fouling-driven parameters that govern operational expenses.

Notably, although the mechanistic insights presented here are derived from urine, they illuminate a methodological principle with broader implications: reliable prediction of BMED performance requires synthetic formulations that preserve the core intermolecular interactions of the target stream. This understanding suggests that when treating other complex waste streams (e.g., from food processing or agricultural operations) where membrane performance depends on multi-constituent interplay, adopting component-representative designs becomes essential.

Consequently, we urge a shift in experimental design from the single-solute simplification paradigm to the organic component realism paradigm, systematically considering the inherent molecular weight distribution, functional group diversity, and cross-scale interaction characteristics in wastewater. Future research should focus on developing hierarchical synthetic formulations that emulate the dynamic complexity of actual wastewater, enabling robust validation of fouling mechanisms and scalability assessments for BMED and related technologies. By bridging the gap between lab-scale idealization and industrial reality, this framework will accelerate the deployment of resilient, resource-recovering wastewater technologies, ensuring their viability in global water sustainability efforts.

## Methods

### Materials and chemical reagents

All chemicals (analytical reagent grade) used in this study were obtained from Macklin and Aladdin. Three representative synthetic urine formulations were prepared based on the composition and concentrations found in actual urine, ranging from simple inorganic ion solutions to complex mixtures of urea, small organic molecules, and large proteins (Supplementary Table 2). Three sets of AEM and four sets of BPM in the BMED were provided by Lanran Co., China, while three sets of CEM were purchased from Astom Co., Japan. Additionally, a single ePTFE-based HFM filament was provided by Hanchen Co., China. Deionized (DI) water was used throughout the study.

### Fabrication of the BMED system

The BMED is a custom-designed electrodialysis unit consisting of three repeated four-chamber sections: the acid, base, feed, and electrode chambers, each separated by 1 mm thick spacers. Each membrane has an effective area of 55 cm², and the electrodes are titanium alloy coated with ruthenium and iridium. To enhance nutrient recovery from urine, an HFM was integrated into the base chamber to capture volatilized ammonia nitrogen. The shell side of the HFM was exposed to the base solution and ammonia gas, while its lumen, with an effective area of 500 cm², was connected to the absorption chamber, circulating $H_2SO_4$.

### Operation of the BMED system

Initially, the acid and base chambers were filled with DI water, and 4 wt.% NaOH was used as the electrolyte, with urine introduced into the feed chamber. Each chamber contained 0.15 L of the initial solution. A peristaltic pump (JIHPUMP, China) circulated the solution through the membrane at a flow rate of 0.4 L min⁻¹. A regulated DC power supply (ITECH, China) applied a constant voltage of 15 V, operating in continuous cycling mode for nutrient separation and recovery from urine. The batches were processed for 2 hours, taking both energy consumption and performance into account. During the process, real-time monitoring was conducted to measure the pH, conductivity, and solution volume in each chamber.

### Analytical methods

Urea concentrations were determined by mixing 1 mL of sample with 2 mL of acid-ferric and 1 mL of diacetylmonoxime-thiosemicarbazide

solutions, followed by heating at 100 °C for 20 min and spectro-photometric measurement at 525 nm[66]. The anions (Cl⁻, $SO_4^{2-}$, and $PO_4^{3-}$) and cations ($Na^+$, $K^+$, $NH_4^+$, $Mg^{2+}$, and $Ca^{2+}$) were detected using an ion chromatograph (ICS-1100, Thermo, USA). Surface analysis of fouled membranes was conducted using a combination of non-destructive, surface-sensitive techniques to preserve the integrity of the foulant-membrane interface and to enable cross-validation of chemical and morphological data. Surface morphology and elemental composition of the fouled membrane were characterized by SEM (MIRA LMS, Tescan, Czech Republic) and EDS (X-MaxN, Oxford, UK). ATR-FTIR (Nicolet iS20, Thermo, USA) and XPS (K-Alpha, Thermo, USA) were utilized to analyze the surface chemical compositions of different membranes. For XPS data analysis, the spectra were cali-brated to C $1s$ (284.8 eV) and fitted using a Shirley background and mixed Gaussian–Lorentzian function. Doublets were constrained by fixed spin-orbit splitting and area ratios. The contact angle and surface free energy of the membrane were measured using a Contact Angle Goniometer (SDC 200S, SINDIN, China).

## Calculations of the BMED system performance evaluation

Desalination efficiency serves as a core metric for evaluating the per-formance of BMED system. The recovery efficiency of individual nutrients represents a key parameter for assessing the efficacy of BMED system in achieving resource recovery from urine treatment; importantly, the performance decay rate stands as a critical determi-nant for evaluating the long-term operational sustainability and relia-bility of such systems.

The desalination efficiency (DE) of urine was calculated using Eq. 1.

$$DE = \frac{\sigma_0 - \sigma_t}{\sigma_0} \times 100\% \tag{1}$$

where $\sigma_O$ and $\sigma_t$ are the conductivity of urine in the feed chamber at times 0 and $t$ (min).

The recovery efficiencies of each nutrient were calculated according to Eq. 2.

$$R = \frac{C_t^a}{C_0^a} \times 100\% \tag{2}$$

where $R$ denotes the recovery efficiency of each nutrient; $C_t^a$ (mg $L^{-1}$) represents the concentration of each nutrient in the recovery chamber —specifically, $NH_4^+$ in the absorption chamber, urea in the feed chamber, $PO_4^{3-}$ in both acid and feed chambers, and $K^+$ in both base and feed chambers—at time $t$ (min). And $C_O^a$ (mg $L^{-1}$) corresponds to the initial concentration of each nutrient in urine.

The performance decay ratio ($P$) of the BMED system was deter-mined using Eq. 3.

$$P = \frac{P_0 - P_n}{P_n} \times 100\% \tag{3}$$

where $P_0$ denotes the initial performance of BMED system in urine treatment (specifically, desalination efficiency), while $P_n$ represents the performance observed during the nth batch of BMED processing.

Notably, the nutrient recovery decay rate ($N$) was calculated to evaluate the decrease in recovery efficiency over consecutive batches, as shown in Eq. (4).

$$N = \frac{N_0 - N_n}{N_0} \times 100\% \tag{4}$$

where $N_0$ and $N_n$ represent the nutrient recovery content (e.g., nitro-gen or phosphorus) in the first batch and the nth, respectively.

## Theoretical calculation of surface free energy

Surface free energy reflects the intermolecular forces at a material's surface and is crucial in determining interactions between membrane materials and pollutants, although it cannot be directly measured. In this study, surface free energy was estimated through contact angle measurements using liquids with known surface free energies (water and diiodomethane), applying the Owens-Wendt model. This model decomposes the membrane's surface free energy and the liquid's surface tension into two components: dispersive (d) and polar (p)[67], as described in Eqs. 5 and 6.

$$\gamma_S = \gamma_S^d + \gamma_S^p \tag{5}$$

$$\gamma_L = \gamma_L^d + \gamma_L^p \tag{6}$$

where $\gamma_S$ represents the total free energy of the membrane, with $\gamma_S^d$ and $\gamma_S^p$ denoting the dispersive and polar components, respectively; $\gamma_L$ is the surface tension of the liquid, while $\gamma_L^d$ and $\gamma_L^p$ correspond to the dispersive and polar components of the liquid's surface free energy.

The model was derived from the Young's equation and the Good-van Oss theory (Eq. 7):

$$\gamma_L(1 + \cos\theta) = 2\sqrt{\gamma_S^d \times \gamma_L^d} + 2\sqrt{\gamma_S^p \times \gamma_L^p} \tag{7}$$

In this study, polar water and nonpolar diiodomethane were selected as titration liquids, with contact angles denoted as $\theta_1$ and $\theta_2$, respectively, as described in Eqs. 8 and 9.

$$\gamma_{L1}(1 + \cos\theta_1) = 2\sqrt{\gamma_S^d \times \gamma_{L1}^d} + 2\sqrt{\gamma_S^p \times \gamma_{L1}^p} \tag{8}$$

$$\gamma_{L2}(1 + \cos\theta_2) = 2\sqrt{\gamma_S^d \times \gamma_{L2}^d} + 2\sqrt{\gamma_S^p \times \gamma_{L2}^p} \tag{9}$$

For polar water, $\gamma_{L1} = 72.8$ mN $m^{-1}$, $\gamma_{L1}^d = 21.8$ mN $m^{-1}$, and $\gamma_{L1}^p = 51.0$ mN $m^{-1}$, while for nonpolar diiodomethane, $\gamma_{L2} = 50.8$ mN $m^{-1}$, $\gamma_{L2}^d = 50.4$ mN $m^{-1}$, and $\gamma_{L2}^p = 0$ mN $m^{-1}$. Solving the two equations simultaneously gives $\gamma_S^d$ and $\gamma_S^p$, with the $\gamma_S$ equal to their sum.

## Molecular dynamics simulation

The interactions between ions and the fragments (with functional groups) of AEM and CEM were simulated in the FORCITE module and COMPASS II force fields in Material Studio 2019[68,69]. In the simulation process, solution-CEM-solution (urea/urea mixture) and solution-AEM-solution (urea/urea mixed) were constructed. All partial atomic char-ges were defined using the adopted force field. A simulation box with dimensions of 28.28 Å × 28.28 Å × 74.44 Å was created. These four systems were geometrically optimized, followed by model construc-tion. With the optimized structure, all calculations were conducted in the NPT and NVT ensembles at 298 K, with a time step of 1 fs and a total simulation time of 600 ps, during which simulation trajectories were recorded every 1000 steps. The Ewald scheme and the atomic trun-cation method were used to handle electrostatic and van der Waals interactions, respectively. Data were collected in the NVT ensemble. The radial distribution function (RDF) was used to describe the inter-actions between ion exchange membranes and solution contents (urea, ions, uric acid, and creatinine).

## Reporting summary

Further information on research design is available in the Nature Portfolio Reporting Summary linked to this article.

## Data availability

The data supporting the findings of this study are available as supplementary materials. Source Data file has been deposited in Figshare under accession code (https://doi.org/10.6084/m9.figshare.29244623[70]).

## Code availability

Code used to perform ion transport modeling is provided in the linked repository. (https://doi.org/10.6084/m9.figshare.29244623[70]).

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

## Acknowledgements

This work was supported by the National Natural Science Foundation of China (52570058, Y.L.; 42307111, M.-Y.Z.; 52300179, D.W.; 52300077, L.-J.X.), the Key Projects for Foreign Cooperation of Bureau of International Cooperation, Chinese Academy of Sciences (309GJHZ2023003MI, Y.L.), the Natural Science Foundation of Chongqing (CSTB2024NSCQ-MSX0897, S.-J.H.; CSTB2025NSCQ-GPX0563, M.-Y.Z.; CSTB2025NSCQ-GPX0562, L.Z.), and the Foundation of Chongqing Water Resources Bureau (202409, L.Z.). We also thank the workers in Scientific Compass (www.shiyanjia.com) for the support of SEM, FT-IR and XPS tests.

## Author contributions

H.-R.Y. and Y.L. conceived the idea. Y.L. supervised the project. H.-R.Y. wrote the manuscript with input from all other authors. S.-J.H., M.-Y.Z., D.W., L.Z., Q.-H.S., Z.-H.Y., L.-J.Z., X. Z., Y.C., H. L., and L.-J.X. assisted with writing-review and editing and data curation. All authors contributed to the discussion of the work.

## Competing interests

The authors declare no competing interests.

## Additional information

**Supplementary information** The online version contains
supplementary material available at

Lin-Ji Xu or Yuan Liu.

**Peer review information** *Nature Communications* thanks the anon-
ymous reviewers for their contribution to the peer review of this work. A
peer review file is available.

