## [Transparent Peer Review file · Nature Communications]

Synthetic urine oversimplification results in misleading membrane fouling mechanisms in bipolar membrane electro dialysis

Corresponding Author: Professor Yuan Liu

Version 0:

Reviewer comments:

Reviewer #1

(Remarks to the Author)

In this work, Yang et al. pointed out that oversimplified wastewater composition, which has been used quite often in previous studies focusing on membrane fouling, can mislead the result. I totally support their view. They demonstrated that this was the case when oversimplified synthetic urine was tested with an electro dialysis system: a very simple composition caused more severe fouling than did more complex compositions. It is a very interesting finding, but it is still dangerous to generalize the obtained results for various membrane processes. I suggest the authors to change the title of this work to avoid such misleading. I feel that this is not the case with most of pressure-driven membrane processes.

They carried out a cost analysis in this work. I do not see the rational and the necessity for it, as they still tested simplified urines in this study. To what extent do they think did the "complex urine" such as Group C represent the real urine? They demonstrated that the very simplified urine (Group A) exhibited more severe fouling than Groups B/C. However, this does not mean Group A still exhibits more severe fouling than does the real urine.

The fundamental information, pH of the solutions measured during the tests, is not shown in the current manuscript. This certainly lowers the quality of this work. Discussion in this work should be made with pH.

L134-140

They need to justify the use of creatinine, uric acid and BSA much more carefully. Are they really dominant metabolites and macromolecular organics in the urine? Particularly, the use of BSA is questionable. BSA does not necessarily represent proteins in the real urine.

L161

There is no data by which we believe that rapid transmembrane migration occurred.

Figs 2 and 3

Quantitative discussion on the basis of EDS and/or FTIR is dangerous. They should have scraped the deposits and analyzed them directly.

L369

How can it be claimed "significantly higher"?

L395

Are there any possibilities of formation of MAP? This could be confirmed by sampling the deposits.

Reviewer #2

(Remarks to the Author)

In the manuscript "Why oversimplified wastewater composition undermines experimental integrity and biases membrane fouling interpretation", a critical gap in wastewater treatment research by demonstrating how oversimplified synthetic urine formulations distort membrane fouling mechanisms and economic assessments in BMED systems was addressed.

However, before its publication some amendments have to be done. Authors must work on the following points:

1. The topic is not focused and the scope is too broad. Suggest similar modifications: "Oversimplified Wastewater Composition Compromises Experimental Integrity and Biases Membrane Fouling Interpretation in Electrodialysis Systems."
2. While focused on urine, the title and abstract broadly implicate "wastewater." Clarify whether conclusions extend to other waste streams (e.g., municipal/industrial). If not, modify claims to avoid overgeneralization (e.g., replace "wastewater" with "source-separated urine" in key statements).
3. Abstract lacks quantitative conclusions. Add quantitative key findings (e.g., "29.3% lower decay rate in multi-component systems") to highlight impact.
4. Introduction lacks sufficient emphasis on how this work fundamentally advances beyond prior literature. Explicitly contrast findings with existing studies to highlight novelty (e.g., the discovery of antagonistic organic interactions that stabilize BMED performance).
5. The hierarchical formulation design (Groups A–C) effectively bridges lab-industry gaps. Yet, the rationale for selecting specific metabolites (creatinine, uric acid, BSA) needs strengthening. Include quantitative data from actual urine (e.g., concentration ranges in Supplementary Table 1) to justify their inclusion and representativeness.
6. Cost analysis is based on lab-scale operations. Projections for industrial-scale viability require sensitivity analysis of key variables (e.g., membrane lifespan under real-world cleaning protocols, fluctuations in energy/chemical costs). Discuss scalability limitations (e.g., electrode degradation, flow distribution challenges).
7. SEM images (Fig. 2a/3a): Include scale bars for all micrographs. Label deposits (e.g., "urea aggregate," "Ca²⁺-BSA co-deposit").
8. Group D (inorganic-only): Clarify why Mg²⁺ scaling occurred on CEM while Ca²⁺ migrated to BPM. Elaborate on the role of ionic radius/polarizability beyond citing references.
9. Relate cost deviations (15.9% cleaning cost overestimation) to industry implications in Discussion.
10. Add error bars to multiple quantitative data graphs to address statistical significance.

Version 1:

Reviewer comments:

Reviewer #1

(Remarks to the Author)

All of the concerns raised by this reviewer has been adequately addressed.

Reviewer #2

(Remarks to the Author)

The author has addressed all the concerns and suggestions of the reviewers, and this version can be accepted for publication.

Point-to-point reply

We are sincerely grateful to the editors and reviewers for their thorough review and insightful comments, which have been instrumental in enhancing the rigor and clarity of our manuscript. All suggestions have been incorporated into the revised version. Our point-by-point responses and the associated revisions are outlined below.

Response Letter to Reviewer 1

Dear Reviewer,

Thank you for reviewing our paper. The paper was modified according to your valuable suggestions and comments.

1. In this work, Yang et al. pointed out that oversimplified wastewater composition, which has been used quite often in previous studies focusing on membrane fouling, can mislead the result. I totally support their view. They demonstrated that this was the case when oversimplified synthetic urine was tested with an electro dialysis system: a very simple composition caused more severe fouling than did more complex compositions. It is a very interesting finding, but it is still dangerous to generalize the obtained results for various membrane processes. I suggest the authors to change the title of this work to avoid such misleading. I feel that this is not the case with most of pressure-driven membrane processes.

Response

We thank the reviewer for their positive feedback on our core premise and their valuable suggestion to refine the scope of our work. We agree that our conclusions should be carefully framed to avoid overgeneralization across membrane processes. To prevent any potential misunderstanding, we have revised the title of the manuscript to "The oversimplification pitfall: How synthetic urine misleads mechanistic understanding of membrane fouling in bipolar membrane electro dialysis" and have carefully refined the language throughout the manuscript—particularly in the abstract and discussion—to consistently frame our conclusions within the context of electro dialysis. We hope that these explanations adequately address your concerns.

We appreciate your valuable suggestions, which have helped us improve the clarity and professionalism of our manuscript.

Revised as: Title (line 1-3), line 98-101, line 357-359: and line 515-518.

Title (Line 1-3): The oversimplification pitfall: How synthetic urine misleads mechanistic understanding of membrane fouling in bipolar membrane electro dialysis

Line 98-101: Consequently, it creates a fundamental knowledge gap: How do complex component synergies—routinely ignored in oversimplified synthetic formulations—impact treatment performance and obscure experimental interpretability of performance-limiting mechanisms in BMED systems?

Line 357-359: These differences underscored the limitations of oversimplified wastewater component models for BMED systems, as their lab-level data failed to guide practical industrial applications.

Line 515-518: This study highlights a critical yet overlooked limitation in membrane-based source-separated urine treatment research, demonstrating using the case of BMED systems that oversimplified synthetic compositions fundamentally distort membrane fouling mechanisms and compromise the translational validity of laboratory findings.

2. They carried out a cost analysis in this work. I do not see the rationale and the necessity for it, as they still tested simplified urines in this study. To what extent do they think did the “complex urine” such as Group C represent the real urine? They demonstrated that the very simplified urine (Group A) exhibited more severe fouling than Groups B/C. However, this does not mean Group A still exhibits more severe fouling than does the real urine.

Response

We thank the reviewer for this insightful comment, which raises important questions regarding the representativeness of our synthetic formulations and the rationale for the cost analysis. We agree that our most complex formulation (Group C) remains a model system and does not fully capture the complete complexity of real urine. In revising our manuscript, we have sought to better articulate the intent behind our experimental design and the economic assessment.

The primary objective of the cost analysis was not to provide a precise industrial cost forecast, but rather to serve as a proof-of-concept demonstration of how compositional oversimplification can systematically bias economic evaluations. We have clarified this point in the text (Lines 459-463). To address the reviewer's specific concern about representativeness, we have now explicitly stated that Group C was designed to

incorporate key organic constituent categories—urea, small-molecule metabolites (creatinine, uric acid), and a model protein (BSA)—that are universally present in real urine and are known to critically influence interfacial processes. Our molecular dynamics simulations and experimental characterizations confirm that these components actively govern the fouling mechanisms under study, making Group C a more physiologically relevant model than the highly simplified formulations commonly used in the literature.

Furthermore, we have reframed the discussion to focus squarely on the comparative analysis between the synthetic formulations that were experimentally tested. The monotonic improvement in system stability and the corresponding reduction in projected costs from Group A to Group C demonstrates that the choice of synthetic formulation itself fundamentally alters the perceived performance and economic outlook. This internal comparison provides a direct, quantitative measure of how oversimplification (as in Group A) can bias experimental outcomes and subsequent engineering assessments compared to a more representative model (as in Group C). The cost overestimation identified with Group A data quantifies the risk of misjudgment inherent in selecting an oversimplified model over a more complex alternative. This reinforces our study's primary contribution: to highlight that the choice of lab-scale model composition is a critical methodological factor that can predetermine technical and economic conclusions.

We hope that these revisions, which include a more cautious interpretation of our cost analysis and a clearer justification of our model formulations, adequately address the reviewer's concerns. The feedback has been invaluable in helping us refine the narrative of our study and better highlight its central message: that overlooking key wastewater constituents can lead to both technical and economic misjudgments in process evaluation.

Revised as: Line 459-463, line 547-554 and new references.

Line 459-463: To demonstrate how formulation simplification can systematically bias such assessments, we performed a proof-of-concept cost-revenue analysis building on prior research and based on our lab-scale data (Fig. 4a and Supplementary Table 3)^{17,57,58}. This analysis is not intended as a precise industrial forecast, but to quantify the economic risks of relying on oversimplified compositions.

Line 547-554: It is crucial to note that this cost analysis, while revealing a systematic bias, is derived from lab-scale operations. To assess the robustness of our finding—that simplified formulations overestimate costs—we first dissected the cost structure. As synthesized in Supplementary Fig. 12, the complex formulation (Group C) achieves a lower total cost primarily by reducing the costs associated with membrane replacement and cleaning (both fixed and operational costs). The economic advantage of Group C lies in its more stable operation, which persists even though it incurs marginally higher energy costs due to longer runtime.

Reference:

- 17 Yang, H.-R. *et al.* Advanced electrochemical membrane technologies for near-complete resource recovery and zero-discharge of urine: Performance optimization and evaluation. *Water Research* **263**, 122175 (2024).
- 57 Kargol, A. K., Burrell, S. R., Chakraborty, I. & Gough, H. L. Synthetic wastewater prepared from readily available materials: Characteristics and economics. *PLoS Water* **2**, e0000178 (2023).
- 58 Roibás-Rozas, A., del Oso, M. S., Posada, J. A., Mosquera-Corral, A. & Hospido, A. A circular economy strategy for valorizing industrial saline wastewaters: Techno-economics and environmental impacts. *Chemical Engineering Journal* **472**, 144819 (2023).

3. The fundamental information, pH of the solutions measured during the tests, is not shown in the current manuscript. This certainly lowers the quality of this work. Discussion in this work should be made with pH.

Response

We thank the reviewer for this insightful comment, which helps strengthen our argument. We agree that the pH is a fundamental parameter governing the BMED process and the discussed fouling mechanisms, and its omission was a significant oversight.

In response, we have now included the pH data for all key chambers (acid, base, and feed) across the operational batches as Supplementary Fig. 1. We have chosen to present

this dataset in the Supplementary Information to maintain the focus of the main text figures on the primary performance metrics (e.g., conductivity, nutrient recovery efficiency, and fouling characterization) that directly underpin the central thesis of formulation simplification-induced bias. However, recognizing its critical role in interpreting the process stability and fouling mechanisms, we have fully integrated the discussion of the pH data into the main text. Furthermore, we have revised the manuscript to integrate the discussion of pH where it is most relevant, using the data to explain the stable process driving force and the serendipitous suppression of urea hydrolysis.

We are grateful for this constructive feedback, which has significantly improved our manuscript.

Revised as: Supplementary Fig. 1, line 161-171 and new references.

Supplementary Fig. 1 | pH in BMED chambers during urine treatment over 7 batches: (a)

acid, (b) base, and (c) feed chambers.

Line 161-171: Despite the differential performance decay, the core functionality of the BMED process remained relatively stable, as reflected in the consistent pH evolution of the key chambers (Supplementary Fig. 1). Both the acid and base chambers established their respective strongly acidic (pH < 2.5) and alkaline (pH > 10.5) environments within the first 5 minutes of operation and maintained them consistently throughout the experiments, providing a stable driving force for nutrient recovery and ammonia stripping. The feed chamber pH remained stable at approximately 3, a consequence of inherent proton leakage in the BMED process²². This phenomenon, however, conferred a critical advantage by serendipitously suppressing urea hydrolysis, thereby obviating the need for exogenous acid addition and significantly facilitating the storage and transportation of urine resources²³.

Reference:

- 22 Foo, Z. H., Lee, T. R., Wegmueller, J. M., Heath, S. M. & Lienhard, J. H. Toward a circular lithium economy with electrodialysis: upcycling spent battery leachates with selective and bipolar ion-exchange membranes. *Environmental Science & Technology* **58**, 19486-19500 (2024).
- 23 Wu, H. *et al.* Nutrient recovery from urine: Urea adsorption onto biochar integrated with Na-chabazite as urease inhibitor. *Resources, Conservation and Recycling* **212**, 107955 (2025).

L134-140

4. They need to justify the use of creatinine, uric acid and BSA much more carefully. Are they really dominant metabolites and macromolecular organics in the urine? Particularly, the use of BSA is questionable. BSA does not necessarily represent proteins in the real urine.

Response

We thank the reviewer for this important question regarding our selection of organic components. In response, we have strengthened the justification in the manuscript. The

small organic metabolites, creatinine and uric acid, were selected because they are consistently identified in literature as core organic constituents of human urine and are frequently employed in studies seeking to enhance the realism of synthetic formulations. For the macromolecular component, BSA was chosen as a representative model protein for several key reasons. Firstly, from a practical standpoint, obtaining a stable and consistent supply of purified human urinary proteins (which encompass a wide range, including albumin, uromodulin, and others) for controlled laboratory experiments is challenging and cost-prohibitive. Secondly, and most importantly, BSA is a scientifically justified surrogate. Extensive literature demonstrates its high structural and functional similarity to Human Serum Albumin (HSA), a key proteinaceous component present in urine. As such, BSA serves as an excellent model for the albumin fraction of urinary protein. Finally, the use of BSA as a representative proteinaceous foulant is a well-established and widely accepted paradigm in fundamental membrane fouling studies. This allows for a controlled, mechanistic investigation of macromolecular interactions (e.g., adsorption, aggregation) that are relevant to the broader class of proteins in urine, providing a critical and reproducible foundation for our hierarchical experimental design. Our hierarchical design (Groups A–C) is intended to systematically bridge the complexity gap by introducing these key, often-overlooked organic categories, thereby providing a more robust platform for analyzing multi-component interactions than single-solute simplifications.

We believe these additions not only address your concerns but also set the stage for more realistic and applicable future research. Thank you for the opportunity to enhance our work based on your constructive feedback.

Revised as: Line 145-152 and new references.

Line 145-152: Group B expands this by incorporating core small organic metabolites—creatinine and uric acid—which are not only universally present in actual urine but are also consistently employed in rigorous studies to enhance the representativeness of synthetic urine formulations^{5,16,17}. Group C further introduces macromolecular organics, using Bovine Serum Albumin (BSA) as a well-established model protein. This choice is justified by its high structural and functional similarity to human serum albumin, a key proteinaceous

component in urine, allowing for a controlled investigation of macromolecular fouling mechanisms relevant to real systems¹⁹⁻²¹.

Reference:

- 5 Li, Y. *et al.* Bipolar membrane electrodialysis for ammonia recovery from synthetic urine: experiments, modeling, and performance analysis. *Environmental Science & Technology* **55**, 14886-14896 (2021).
- 16 Yuan, X., Liu, J., Han, C., Li, Y. & Feng, Y. Simultaneous nutrient-energy recovery from source-separated urine based on bioelectrically enhanced bipolar membrane-driven in-situ alkali production coupling with gas-permeable membrane system. *Chemical Engineering Journal* **431**, 134161 (2022).
- 17 Yang, H.-R. *et al.* Advanced electrochemical membrane technologies for near-complete resource recovery and zero-discharge of urine: Performance optimization and evaluation. *Water Research* **263**, 122175 (2024).
- 19 Li, J. *et al.* A two-dimensional fluorescence and chemiluminescence orthogonal probe for discriminating and quantifying similar proteins. *Chemical Science* **16**, 3228-3237 (2025).
- 20 Gupta, R. & Paul, K. A fluorescent "Turn-ON" probe with rapid and differential response to HSA and BSA: quantitative detection of HSA in urine. *Journal of Materials Chemistry B* **12**, 9037-9049 (2024).
- 21 Yang, X. *et al.* Tetraphenylethylene-indole as a novel fluorescent probe for selective and sensitive detection of human serum albumin (HSA) in biological matrices and monitoring of HSA purity and degradation. *Talanta* **286**, 127471 (2025).

L161

5. There is no data by which we believe that rapid transmembrane migration occurred.

Response

Thank you for this insightful comment. We agree that the term "rapid" overstepped what could be directly supported by our kinetic data. In response, we have revised the text to use the term "stable" to describe NH_4^+ migration, which is accurately reflected by its

consistent recovery performance throughout the batches (Fig. 1d) and does not imply a specific rate.

We appreciate your feedback in strengthening the precision of our wording.

Revised as: Line 182-184.

Line 182-184: As shown in Fig. 1d, NH_4^+ recovery remained largely unaffected by performance decay due to its stable transmembrane migration, a finding consistent with results from previous studies^{17,26}.

Figs 2 and 3

6. Quantitative discussion on the basis of EDS and/or FTIR is dangerous. They should have scraped the deposits and analyzed them directly.

Response

We are grateful to the reviewer for raising this important point regarding analytical rigor. We fully acknowledge that mechanical scraping of deposits, followed by bulk analysis, is a powerful and direct method for foulant characterization. Under circumstances where a stable, scrapable foulant layer can be obtained, it undoubtedly provides invaluable compositional insights.

However, within the specific context of bipolar membrane electrodialysis (BMED) operated under constant voltage, this approach is often precluded by fundamental process constraints. As noted in previous studies¹⁻³, severe fouling in electromembrane processes primarily manifests as a drastic increase in electrical resistance rather than the formation of a thick, physical cake. This resistance surge leads to excessive Joule heating and risks irreversible membrane damage long before a mechanically stable, scrapable layer is formed. Indeed, as shown in the Fig. R1, which compares the initial membranes (a: pristine CEM; c: pristine AEM) with those subjected to seven batches of BMED operation under Group A conditions (b: fouled CEM; d: fouled AEM), no macroscopic or mechanically scrapable foulant layer is visually discernible on the fouled membranes, despite the severe performance decay recorded (Fig. 1c). This operational limitation, which necessitates alternative characterization strategies, is a recognized challenge in the electromembrane

field^{4,5}.

Fig. R1. Surface comparison of initial and fouled membranes. (a) Pristine CEM; (b) Fouled CEM from Group A after seven batches; (c) Pristine AEM; (d) Fouled AEM from Group A after seven batches. All membranes exhibit uniform surfaces without macroscopic foulant layers, consistent with the typical manifestation of electro dialysis fouling as a thin, adherent film that elevates electrical resistance.

Therefore, in alignment with established practices for such systems, our experimental strategy was designed around a suite of non-destructive, surface-sensitive techniques (SEM-EDS, FTIR, XPS, XRD). This multi-methodological approach is widely adopted to preserve the integrity of the fragile foulant-membrane interface and to enable cross-validation of data^{4,5}. In direct response to the reviewer's valid concern about over-interpretation, we have revised the manuscript to explicitly emphasize the semi-quantitative nature of the EDS analysis and to clarify that all mechanistic conclusions are drawn from the convergence of evidence provided by these complementary techniques, rather than relying on any single method. We believe this in-situ, multi-technique framework not only addresses a technical constraint but also provides a robust, multi-faceted understanding of the fouling layer.

We thank the reviewer again for pushing us to articulate these methodological considerations more clearly.

Reference:

- 1 Titorova, V. *et al.* How bulk and surface properties of sulfonated cation-exchange membranes response to their exposure to electric current during electro dialysis of a Ca²⁺ containing solution. *Journal of Membrane Science* **644**, 120149 (2022).

- 2 Meng, J. *et al.* Membrane fouling during nutrient recovery from digestate using electro dialysis: impacts of the molecular size of dissolved organic matter. *Journal of Membrane Science* **685**, 121974 (2023).
- 3 Ghalloussi, R. *et al.* Ageing of ion-exchange membranes in electro dialysis: A structural and physicochemical investigation. *Journal of Membrane Science* **436**, 68-78 (2013).
- 4 Zhang, S. *et al.* Insight into membrane fouling mechanism and cleaning strategy during selective electro dialysis for metal ions removal from ionic liquid aqueous solutions. *Journal of Environmental Chemical Engineering* **11**, 109246 (2023).
- 5 Xia, Q. *et al.* Interaction mechanisms between fouling and chemical cleaning on the ageing behavior of ion-exchange membranes during electro dialysis treatment of flue gas desulfurization wastewater. *Water Research* **271**, 122897 (2025).

Revised as: Line 217-220, line 368-372, line 413-417 and line 616-619.

Line 217-220: Scanning electron microscopy coupled with energy-dispersive X-ray spectroscopy (SEM-EDS) investigates the spatial distribution and semi-quantitative elemental composition of the fouling layer of the anion-exchange membrane (AEM) exposed to various synthetic urine formulations.

Line 368-372: EDS analysis further indicated a significant enrichment of hardness ions (Ca^{2+} , Mg^{2+}) in CEM deposits (Fig. 3b), with Ca^{2+} concentrations ranging from 0.5-3 wt% and Mg^{2+} from 1-6 wt% across Groups A, B, and C, providing semi-quantitative evidence that these ions were critical mediators of foulant formation.

Line 413-417: The consistent findings from EDS, FTIR, and XPS collectively suggest that ionic electrostatic interactions and metal-ligand coordination jointly mediated the adsorption of Ca^{2+} and Mg^{2+} on CEM, which in turn promoted urea aggregation via two pathways: (1) direct coordination between metal ions and carbonyl/amino groups of urea, and (2) intermolecular bridging by charged ions to stabilize urea aggregates.

Line 616-619: Surface analysis of fouled membranes was conducted using a combination of non-destructive, surface-sensitive techniques to preserve the integrity of the foulant-membrane interface and to enable cross-validation of chemical and morphological data.

L369

7. How can it be claimed "significantly higher"?

Response

Thank you for your valuable comments. We agree that the term "significantly higher" was inappropriate without supporting statistical testing. We have revised the manuscript to replace this and similar subjective terms with more accurate, descriptive language that reflects the observed differences in intensity or concentration without implying statistical significance. We appreciate your feedback, which has enhanced the precision of our scientific language.

Revised as: Line 395-400.

Line 395-400: FTIR spectroscopy demonstrated that the absorption intensities of C=O ($\sim 1650\text{ cm}^{-1}$) and N-H stretching ($\sim 3400\text{ cm}^{-1}$) vibrations on the CEM surface in Groups A-C were readily detectable and more intense than those of the pristine membrane (Fig. 3c), indicating hydrogen bond formation between the carbonyl/amino groups of urea and the oxygen atoms of sulfonate groups ($-\text{SO}_3^-$), which are inherent to the CEM.

L395

8. Are there any possibilities of formation of magnesium ammonium phosphate (MAP)?

This could be confirmed by sampling the deposits.

Response

We are grateful to the reviewer for raising this pertinent point regarding the potential formation of magnesium ammonium phosphate (struvite, MAP), which is indeed an important consideration in urine treatment processes. In response, we have supplemented the elemental mapping analysis of the fouled membranes to include nitrogen distribution.

As now detailed in the revised Supplementary Note 2 and Supplementary Fig. 9, the scalants identified on the membrane surfaces show a complete absence of nitrogen signal. This finding is critical, as MAP ($\text{MgNH}_4\text{PO}_4 \cdot 6\text{H}_2\text{O}$) necessarily contains nitrogen in its structure; the lack of it in the deposits allows us to definitively exclude MAP as a constituent.

This experimental observation is entirely consistent with, and further reinforced by, the intrinsic ion-separation mechanism of the bipolar membrane electro dialysis system. In our configuration, phosphate anions are electromigrated through the anion-exchange membranes into the acid chamber, rendering the base chamber—where Mg^{2+} and NH_4^+ are present—effectively free of phosphate. The physical segregation of these essential precursors makes the formation of MAP thermodynamically and kinetically implausible.

Consequently, the scaling must be attributed to species that are locally available. The strong correlation between Mg and O, in the absence of both N and P, on the CEM surface facing the base chamber, together with the highly alkaline conditions, provides coherent and conclusive support for the formation of magnesium hydroxide. We have revised the manuscript and Supplementary Note 2 accordingly to present this integrated argument.

Thank you for this critical question, which was essential for us to robustly justify our identification of the scaling constituents.

Revised as: Line 422-425, Supplementary Note 2 and Supplementary Fig. 9.

Line 422-425: SEM mapping revealed ion-specific deposition patterns: Ca^{2+} forming calcium hydroxide scale on the BPM anion side, while Mg^{2+} forming magnesium hydroxide on the CEM base chamber side (Supplementary Figs. 9 and 10; as detailed in Supplementary Note 2).

Supplementary Note 2: Analysis of hydroxide scaling formation on membrane surfaces

The spatial distribution of inorganic scales on the membrane surfaces, as revealed by SEM-EDS elemental mapping (Supplementary Figs. 9 and 10), provides direct evidence for the formation of distinct hydroxide deposits. The mapping shows a clear spatial correlation between Ca and O on the anion-exchange side of the bipolar membrane (BPM), and a concurrent correlation between Mg and O on the base chamber-side surface of the cation exchange membrane (CEM). Critically, the absence of a nitrogen signal in these deposits definitively excludes nitrogen-containing scales such as struvite ($MgNH_4PO_4 \cdot 6H_2O$, MAP). Furthermore, the system design inherently excludes phosphate from the base chamber, as phosphate anions electromigrate through the anion exchange

membrane (AEM) toward the acid chamber. This elemental co-localization, in the absence of other scaling precursor ions, identifies the primary deposits as $\text{Ca}(\text{OH})_2$ and $\text{Mg}(\text{OH})_2$, respectively.

The divergent scaling locations are a direct result of the differential mobility of Ca^{2+} and Mg^{2+} under the applied electric field. Mg^{2+} , with its high charge density and strong hydration energy (-1922 kJ/mol), interacts more strongly with the CEM surface and possesses lower mobility. This results in its preferential accumulation and rapid precipitation as $\text{Mg}(\text{OH})_2$ upon encountering the high-pH environment at the CEM surface, effectively hindering its further transport. In contrast, the more weakly hydrated Ca^{2+} (hydration energy: -1577 kJ/mol) exhibits greater mobility, facilitating its transit across the CEM^{21,22}. Once in the base chamber, Ca^{2+} is further driven towards the BPM by the electric field, where it encounters the most concentrated OH^- flux and precipitates as $\text{Ca}(\text{OH})_2$ on the BPM's anion-exchange layer.

In conclusion, the scaling pattern is not stochastic but is a deterministic outcome of the system's electrochemistry. The combined evidence from elemental mapping and thermodynamic analysis confirms that the observed deposits are $\text{Ca}(\text{OH})_2$ and $\text{Mg}(\text{OH})_2$, whose formation sites are kinetically controlled by the differential migration of the hardness ions toward a pervasive high-pH sink.

Supplementary Fig. 9 | a Surface mapping analysis after treatment with Group D. BP-AEM: Cathode side of the bipolar membrane, i.e., the OH^- production side and **b** Contamination characteristic of BP-AEM surfaces after seven batches of BMED treatment with different urine components (Groups A to C).

Reference:

- 21 Yang, H.-R. *et al.* Advanced electrochemical membrane technologies for near-complete resource recovery and zero-discharge of urine: Performance optimization and evaluation. *Water Research* **263**, 122175 (2024).
- 22 Raghuvanshi, S. *et al.* Dual control on structure and magnetic properties of Mg ferrite: role of swift heavy ion irradiation. *Journal of Magnetism and Magnetic Materials* **471**, 521-528 (2019).

Response Letter to Reviewer 2

Dear Reviewer,

Thank you for reviewing our paper. The paper was modified according to your valuable suggestions and comments.

In the manuscript "Why oversimplified wastewater composition undermines experimental integrity and biases membrane fouling interpretation", a critical gap in wastewater treatment research by demonstrating how oversimplified synthetic urine formulations distort membrane fouling mechanisms and economic assessments in BMED systems was addressed. However, before its publication some amendments have to be done. Authors must work on the following points:

1. The topic is not focused and the scope is too broad. Suggest similar modifications: "Oversimplified Wastewater Composition Compromises Experimental Integrity and Biases Membrane Fouling Interpretation in Electrodialysis Systems."

Response

We sincerely thank the reviewer for their insightful suggestion to refine our title to better highlight the core concern of oversimplification. Inspired by this excellent advice, we have refined the title to: "The oversimplification pitfall: How synthetic urine misleads mechanistic understanding of membrane fouling in bipolar membrane electro dialysis". We believe this version captures the essence of the reviewer's input while more precisely framing our study within the specific context of synthetic urine and bipolar membrane electro dialysis. Accordingly, we have updated the abstract, introduction, and discussion throughout the manuscript to ensure full alignment with this focused narrative.

We are truly grateful for this suggestion, which has significantly strengthened the presentation of our work.

Revised as: Title (line 1-3), line 98-101, line 357-359: and line 515-518.

Title (line 1-3): The oversimplification pitfall: How synthetic urine misleads mechanistic understanding of membrane fouling in bipolar membrane electro dialysis

Line 98-101: Consequently, it creates a fundamental knowledge gap: How do complex

component synergies—routinely ignored in oversimplified synthetic formulations—impact treatment performance and obscure experimental interpretability of performance-limiting mechanisms in BMED systems?

Line 357-359: These differences underscored the limitations of oversimplified wastewater component models for BMED systems, as their lab-level data failed to guide practical industrial applications.

Line 515-518: This study highlights a critical yet overlooked limitation in membrane-based source-separated urine treatment research, demonstrating using the case of BMED systems that oversimplified synthetic compositions fundamentally distort membrane fouling mechanisms and compromise the translational validity of laboratory findings.

2. While focused on urine, the title and abstract broadly implicate "wastewater." Clarify whether conclusions extend to other waste streams (e.g., municipal/industrial). If not, modify claims to avoid overgeneralization (e.g., replace "wastewater" with "source-separated urine" in key statements).

Response

We thank the reviewer for this important comment, which helped us to clarify the scope of our claims. We recognize that the use of the broad term "wastewater" could cause confusion regarding the direct applicability of our findings. In response, we have revised the manuscript to more precisely articulate the basis and implications of our work.

Specifically, we have replaced overarching uses of "wastewater" with "urine" in the manuscript to accurately reflect our experimental system. More importantly, we have added a clarifying paragraph in the Discussion to explicitly state that while our experimental validation was performed on urine, the methodological insight—that oversimplification neglects critical intermolecular interactions—is a fundamental principle likely applicable to the electrodialysis treatment of other complex aqueous streams. This revision ensures our claims are firmly grounded in our data while justifiably highlighting the broader relevance of our conclusions for the field.

We hope that these explanations adequately address your concerns. Thank you again

for your review and valuable suggestions.

Revised as: Line 46-47, line 205-208 and line 563-569. lines 153, 202, 357, 442, 464, 509, 511.

Line 46-47: Keywords: Experimental integrity, Oversimplified urine compositions, Bipolar membrane electro dialysis, Membrane fouling, Engineering-economic assessment.

Line 205-208: These findings suggested that compositional complexity likely stabilizes BMED performance through interactions among urinary components—a mechanism often overlooked in simplified formulations—that may extend to other complex aqueous systems rich in organics and inorganics.

Line 563-569: Notably, although the mechanistic insights presented here are derived from urine, they illuminate a methodological principle with broader implications: reliable prediction of BMED performance requires synthetic formulations that preserve the core intermolecular interactions of the target stream. This understanding suggests that when treating other complex waste streams (e.g., from food processing or agricultural operations) where membrane performance depends on multi-constituent interplay, adopting component-representative designs becomes essential.

3. Abstract lacks quantitative conclusions. Add quantitative key findings (e.g., "29.3% lower decay rate in multi-component systems") to highlight impact.

Response

We thank the reviewer for this excellent suggestion to enhance the impact of our abstract. We agree that including quantitative key findings will provide readers with a more immediate and concrete understanding of our study's contributions. Following this advice, we have revised the abstract to incorporate the most salient quantitative results, including the 29.3% reduction in performance decay, the 15.9% overestimation of cleaning costs, and key improvements in nutrient recovery efficiency.

Thank you once again for your constructive feedback.

Revised as: Line 35-37 and line 40-41.

Line 35-37: After seven batches, the full-component formulations showed 29.3% less

performance decay and 10–14% higher urea recovery than the simplified formulation.

Line 40-41: Simplification also distorts the engineering-economic assessment, overestimating cleaning costs by 15.9% and underestimating membrane lifespan by 12.5%.

4. Introduction lacks sufficient emphasis on how this work fundamentally advances beyond prior literature. Explicitly contrast findings with existing studies to highlight novelty (e.g., the discovery of antagonistic organic interactions that stabilize BMED performance).

Response

We thank the reviewer for this critical suggestion to better articulate the novel contributions of our work. We agree that a stronger emphasis on the conceptual advance beyond prior literature would enhance the introduction. In response, we have substantially revised the introduction to explicitly contrast the established paradigm with our counterintuitive findings, specifically highlighting the discovery of antagonistic organic interactions that stabilize BMED performance—a phenomenon that challenges the conventional wisdom in membrane fouling. Additionally, we have reinforced this point in the Discussion section to further underscore the novelty of our findings.

Thank you once again for your constructive feedback.

Revised as: Line 102-107, line 530-534 and new references.

Line 102-107: Critically, the prevailing reliance on oversimplified formulations (e.g., containing only urea) creates a systematic blind spot^{5,10,16,17}. This paradigm is fundamentally limited, as it inherently fails to capture the full spectrum of intermolecular interactions—particularly the overlooked antagonistic interactions that, counterintuitively, mitigate rather than exacerbate membrane fouling. This oversight not only obscures key fouling mechanisms but also risks a complete misdiagnosis of fouling risks in real-world systems.

Line 530-534: In contrast to the prevailing focus on synergistic fouling, this work establishes that antagonistic organic interactions decisively stabilize BMED performance. This finding challenges the paradigm that wastewater complexity inherently compromises membrane operation and provides a new mechanistic foundation for predicting fouling in

real-world systems.

Reference:

- 5 Li, Y. *et al.* Bipolar membrane electrodialysis for ammonia recovery from synthetic urine: experiments, modeling, and performance analysis. *Environmental Science & Technology* **55**, 14886-14896 (2021).
- 10 Xu, L. *et al.* Selective recovery of phosphorus from synthetic urine using flow-electrode capacitive deionization (FCDI)-based technology. *ACS ES&T Water* **1**, 175-184 (2020).
- 16 Yuan, X., Liu, J., Han, C., Li, Y. & Feng, Y. Simultaneous nutrient-energy recovery from source-separated urine based on bioelectrically enhanced bipolar membrane-driven in-situ alkali production coupling with gas-permeable membrane system. *Chemical Engineering Journal* **431**, 134161 (2022).
- 17 Yang, H.-R. *et al.* Advanced electrochemical membrane technologies for near-complete resource recovery and zero-discharge of urine: Performance optimization and evaluation. *Water Research* **263**, 122175 (2024).

5. The hierarchical formulation design (Groups A–C) effectively bridges lab-industry gaps. Yet, the rationale for selecting specific metabolites (creatinine, uric acid, BSA) needs strengthening. Include quantitative data from actual urine (e.g., concentration ranges in Supplementary Table 1) to justify their inclusion and representativeness.

Response

We sincerely thank the reviewer for this valuable suggestion. In response, we have strengthened the rationale for the selection of specific metabolites (creatinine, uric acid, BSA) in our synthetic formulations. Specifically, we have added a new Supplementary Table 1 to provide quantitative data on their physiological concentrations and have revised the manuscript text to elaborate on the scientific reasoning behind their inclusion. We are grateful for this insightful comment, which has helped strengthen the methodological foundation of our study.

Revised as: Line 142-152, Supplementary Table 1 and new references.

Line 142-152: To bridge the gap between synthetic and real wastewater, the formulations were designed based on comprehensive analysis of urban urine composition (Supplementary Table 1-2). Group A represents the conventional simplified formulation^{9,17}, containing only dominant inorganic salts (e.g., NaCl, KCl) and urea. Group B expands this by incorporating core small organic metabolites—creatinine and uric acid—which are not only universally present in actual urine but are also consistently employed in rigorous studies to enhance the representativeness of synthetic urine formulations^{5,16,17}. Group C further introduces macromolecular organics, using Bovine Serum Albumin (BSA) as a well-established model protein. This choice is justified by its high structural and functional similarity to human serum albumin, a key proteinaceous component in urine, allowing for a controlled investigation of macromolecular fouling mechanisms relevant to real systems¹⁹⁻²¹.

Supplementary Table 1 | Typical concentration ranges of major components in human urine.

Composition	Normal range in humans
Urea (CH ₄ N ₂ O)	10–35 g/d
Uric acid (C ₅ H ₄ N ₄ O ₃)	<750 mg/d
Creatinine (C ₄ H ₇ N ₃ O)	Males: 955-2936 mg/d Females: 601-1689mg/d
Citrate (C ₆ H ₅ O ₇ ³⁻)	221-1191 mg/d
Sodium (Na ⁺)	41-227 mmol/d
Potassium (K ⁺)	17-77 mmol/d
Ammonium (NH ₄ ⁺)	15-56 mmol/d
Calcium (Ca ²⁺)	Males: < 250 mg/d Females: < 200 mg/d
Magnesium (Mg ²⁺)	51-269 mg/d
Chloride (Cl ⁻)	40-224 mmol/d
Oxalate (C ₂ O ₄ ²⁻)	0.11-0.46 mmol/d

Sulphate (SO ₄ ²⁻)	7-47 mmol/d
Phosphate (PO ₄ ³⁻)	20-50 mmol/d

Note: Data are compiled from reference^{26,27}.

Reference:

- 5 Li, Y. *et al.* Bipolar membrane electrodialysis for ammonia recovery from synthetic urine: experiments, modeling, and performance analysis. *Environmental Science & Technology* **55**, 14886-14896 (2021).
- 16 Yuan, X., Liu, J., Han, C., Li, Y. & Feng, Y. Simultaneous nutrient-energy recovery from source-separated urine based on bioelectrically enhanced bipolar membrane-driven in-situ alkali production coupling with gas-permeable membrane system. *Chemical Engineering Journal* **431**, 134161 (2022).
- 17 Yang, H.-R. *et al.* Advanced electrochemical membrane technologies for near-complete resource recovery and zero-discharge of urine: Performance optimization and evaluation. *Water Research* **263**, 122175 (2024).
- 19 Li, J. *et al.* A two-dimensional fluorescence and chemiluminescence orthogonal probe for discriminating and quantifying similar proteins. *Chemical Science* **16**, 3228-3237 (2025).
- 20 Gupta, R. & Paul, K. A fluorescent "Turn-ON" probe with rapid and differential response to HSA and BSA: quantitative detection of HSA in urine. *Journal of Materials Chemistry B* **12**, 9037-9049 (2024).
- 21 Yang, X. *et al.* Tetraphenylethylene-indole as a novel fluorescent probe for selective and sensitive detection of human serum albumin (HSA) in biological matrices and monitoring of HSA purity and degradation. *Talanta* **286**, 127471 (2025).
- 26 Sarigul, N., Korkmaz, F. & Kurultak, İ. A new artificial urine protocol to better imitate human urine. *Scientific reports* **9**, 20159 (2019).
- 27 Simha, P., Courtney, C. & Randall, D. G. An urgent call for using real human urine in decentralized sanitation research and advancing protocols for preparing synthetic urine. *Frontiers in Environmental Science* **12**, 1367982 (2024).

6. Cost analysis is based on lab-scale operations. Projections for industrial-scale viability require sensitivity analysis of key variables (e.g., membrane lifespan under real-world cleaning protocols, fluctuations in energy/chemical costs). Discuss scalability limitations (e.g., electrode degradation, flow distribution challenges).

Response

We sincerely thank the reviewer for this critical comment, which has significantly strengthened the robustness of our economic discussion. In direct response, we have performed a comprehensive economic sensitivity analysis. The key result is synthesized in the new Supplementary Fig. 12, which visually demonstrates how the superior fouling resistance of the multi-component formulation (Group C) translates into tangible economic benefits—primarily through extended membrane lifespan and reduced cleaning frequency—despite a marginally higher energy cost. This analysis, based on industrially-informed parameter variations (Supplementary Table 5), confirms that the cost advantage of the representative formulation is robust. Furthermore, we have revised the Discussion section to explicitly acknowledge the lab-scale nature of our analysis and to incorporate the reviewer's point regarding scalability limitations.

We are grateful for this suggestion, which has enabled us to provide a more balanced and credible perspective on the technological translation of our findings.

Revised as: Line 547-562, Supplementary Table 5 and Supplementary Fig. 12.

Line 547-562: It is crucial to note that this cost analysis, while revealing a systematic bias, is derived from lab-scale operations. To assess the robustness of our finding—that simplified formulations overestimate costs—we first dissected the cost structure. As synthesized in Supplementary Fig. 12, the complex formulation (Group C) achieves a lower total cost primarily by reducing the costs associated with membrane replacement and cleaning (both fixed and operational costs). The economic advantage of Group C lies in its more stable operation, which persists even though it incurs marginally higher energy costs due to longer runtime. We then performed a single-parameter sensitivity analysis on key parameters (e.g., membrane lifespan, energy cost) using industrially-informed variation ranges (Supplementary Table 5)⁶⁵. The result confirms that the economic superiority of

Group C remains robust across all tested scenarios. While a full industrial assessment must consider additional scale-dependent factors such as electrode degradation and flow distribution challenges, our lab-scale model unequivocally demonstrates that compositional simplification is a fundamental source of economic misjudgment by distorting the key fouling-driven parameters that govern operational expenses.

Supplementary Table 5 | Key parameters and assumed ranges for economic sensitivity analysis of urine treatment via BMED.

Parameter	Variation range	Rationale for variation
Membrane lifetime	±50% (2-6 years)	The typical service life of industrial bipolar membranes is 3-5 years, with 6 years being the upper limit under ideal maintenance conditions.
Cleaning interval	Group A: ±16.7% Groups B/C: ±14.3%	Cleaning frequency is affected by the influent pollutant load, and in industrial operation, it typically fluctuates by ±15-20%, depending on the membrane fouling rate.
Energy price	±20%	Energy price fluctuations are often set at ±20% in international industrial contexts, reflecting regional differences and energy market movements.
Chemical cost	±20%	The prices of acid and alkali reagents are influenced by raw material and transportation costs, and industry analysis often adopts a ±20% fluctuation.
Labour cost	±15%	Labour costs are affected by regional minimum wage policies and labour market supply and demand, with the fluctuation range typically being ±10-15%.

Supplementary Fig. 12 | Relative cost comparison of urine treatment via BMED: complex (Group C) versus simplified (Group A) formulation. The chart shows the ratio of each major cost component for Group C relative to Group A (baseline = 1.0). Values were calculated using the cost data in Supplementary Table 3 and parameter ranges in Supplementary Table 5. Although Group C incurs a higher energy cost (ratio > 1.0) due to its longer, stable operation, this is outweighed by substantial savings in membrane replacement, chemical, and labor costs (all ratios < 1.0), which result from its superior fouling resistance and extended cleaning intervals.

Reference:

65 Newnan, D. G., Eschenbach, T. G. & Lavelle, J. P. *Engineering economic analysis*. Vol. 1 (Oxford University Press, 2004).

7. SEM images (Fig. 2a/3a): Include scale bars for all micrographs. Label deposits (e.g., "urea aggregate," "Ca²⁺-BSA co-deposit").

Response

We thank the reviewer for these critical suggestions to improve the clarity of our data. In response, we have added scale bars to all SEM micrographs in Figures 2a and 3a. Furthermore, to precisely identify the foulants without cluttering the images, we have now detailed their chemical composition directly in the figure captions, with each label explicitly linked to the corresponding morphological features in the micrographs and cross-validated by our EDS analysis. We believe this approach enhances clarity while maintaining a clean

presentation.

We appreciate your feedback, which has significantly improved the presentation of our results.

Revised as: Figs. 2a and 3a. Line 252-256 and line 384-387.

Line 252-256: Fig. 2 | AEM fouling in the BMED system during urine nutrient recovery from different groups over seven batches: a SEM morphological analysis (In Group A (urea-only), spherical primary urea aggregates constitute the fouling layer. In Group B (with creatinine/uric acid), the density of urea aggregates is significantly reduced due to competitive inhibition, resulting in a cleaner membrane surface. In Group C (with BSA), a uniform proteinaceous layer predominates, demonstrating steric hindrance.);

Line 384-387: Fig. 3 | CEM fouling in the BMED system during urine nutrient recovery from different groups over seven batches. a SEM morphological analysis (Dominant Ca^{2+} - Mg^{2+} -urea co-deposits in Group A; a heterogeneous composite incorporating creatinine and uric acid in Group B; and foulants superseded by a BSA- Ca^{2+} - Mg^{2+} complex in Group C.);

8. Group D (inorganic-only): Clarify why Mg^{2+} scaling occurred on CEM while Ca^{2+} migrated to BPM. Elaborate on the role of ionic radius/polarizability beyond citing references.

Response

We thank the reviewer for raising this point to clarify the ion-specific behaviors. We have revised the manuscript to provide a more detailed explanation, elaborating on the distinct hydration properties (ionic radius and hydration energy) of Ca^{2+} and Mg^{2+} to account for their different scaling locations.

Thank you once again for your constructive feedback.

Revised as: Line 425-435 and new references.

Line 425-435: This divergence arose from distinct coordination chemistries: Ca^{2+} (ionic radius: 0.99 Å, with lower charge density and higher polarizability^{52,53}) preferentially bound urea's groups as nucleation cores in organic-containing formulations (Groups A-C), forming steric aggregates that anchored ions to CEM. The distinct hydration properties of

Ca²⁺ and Mg²⁺ further governed their migration paths. Mg²⁺ (ionic radius: 0.71 Å), with its higher charge density, possesses a larger hydration shell and higher hydration energy (-1922 kJ/mol for Mg²⁺ vs -1577 kJ/mol for Ca²⁺), which restricted its mobility and enhanced its retention on the CEM^{17,54}. This preference, in turn, conferred greater mobility upon the more weakly hydrated Ca²⁺, facilitating its transit across the CEM and subsequent precipitation at the BPM interface.

Reference:

- 17 Yang, H.-R. *et al.* Advanced electrochemical membrane technologies for near-complete resource recovery and zero-discharge of urine: Performance optimization and evaluation. *Water Research* **263**, 122175 (2024).
- 52 Vrettos, J. S., Stone, D. A. & Brudvig, G. W. Quantifying the ion selectivity of the Ca²⁺ site in photosystem II: evidence for direct involvement of Ca²⁺ in O₂ formation. *Biochemistry* **40**, 7937-7945 (2001).
- 53 Jiao, D., King, C., Grossfield, A., Darden, T. A. & Ren, P. Simulation of Ca²⁺ and Mg²⁺ solvation using polarizable atomic multipole potential. *The journal of physical chemistry B* **110**, 18553-18559 (2006).
- 54 Raghuvanshi, S. *et al.* Dual control on structure and magnetic properties of Mg ferrite: role of swift heavy ion irradiation. *Journal of Magnetism and Magnetic Materials* **471**, 521-528 (2019).

9. Relate cost deviations (15.9% cleaning cost overestimation) to industry implications in Discussion.

Response

We thank the reviewer for this valuable suggestion to strengthen the practical relevance of our findings. We have revised the Discussion section to explicitly connect the quantified cost deviations (e.g., the 15.9% overestimation of cleaning costs) to their potential implications for industrial decision-making and techno-economic assessments.

Thank you once again for your constructive feedback. We believe these revisions enhance the robustness and relevance of our study.

Revised as: Line 544-547.

Line 544-547: When scaled to industrial levels, these systematic errors would cascade into substantial financial miscalculations for full-scale plants, distorting projections of operating expenses, misguiding capital investment decisions, and ultimately misrepresenting the economic competitiveness of the BMED technology.

10. Add error bars to multiple quantitative data graphs to address statistical significance.

Response

We sincerely thank the reviewer for this insightful comment regarding statistical significance. We agree that replicating experiments is a cornerstone of robust scientific practice.

In designing this study, our primary objective was to mechanistically characterize the evolution of membrane fouling and its impact on system performance over an extended, continuous operation. This approach, using a single reactor run through multiple consecutive batches, is particularly suited to capturing the inherent trends and progressive decay mechanisms in electrochemical systems, which can be masked by averaging across independent runs. We acknowledge that this design choice means we cannot provide traditional error bars derived from parallel reactors for the performance data in Fig. 1b-f.

However, we wish to emphasize that the observed performance trends are robustly and consistently supported by multiple, deterministic lines of evidence within the same experimental sequence:

1. Strong correlation with fouling characterization: The performance decay directly correlates with the spatially and chemically resolved formation of foulants, as definitively shown by post-operation analysis (SEM-EDS, FTIR, XPS in Figs. 2 and 3). This provides a mechanistic, causal link to the observed performance drop.
2. Internal consistency of process parameters: Other key parameters monitored throughout the run, such as the stable establishment of pH gradients (Supplementary Fig. 1) and the logical progression of product concentration changes, show a coherent and consistent narrative that aligns perfectly with the performance decay trend.

3. Initial system stability: The high and stable performance in the initial batches confirms the baseline reliability of our experimental setup before significant fouling occurred.

In response to the reviewer's valid point, we have revised the figure captions to explicitly state: "Data points represent measurements from a single, continuous operational sequence for each group."

We fully acknowledge that future work incorporating multiple independent runs would provide valuable statistical validation of absolute performance values. We are grateful for this comment, which has improved the clarity of our manuscript and will directly inform the design of our subsequent validation studies.

Revised as: Line 141.

Line 141: Data points represent measurements from a single, continuous operational sequence for each group.

Point-to-point reply

We are sincerely grateful to the Editor for the meticulous final check and insightful comments regarding the data presentation, which have been instrumental in enhancing the rigor and accuracy of our manuscript. All suggestions have been incorporated into the revised version. Our point-by-point responses and the associated revisions are outlined below.

Response Letter to Editor

Dear Editor,

Thank you for the final technical assessment of our paper. The manuscript and figures were modified strictly according to your requirements regarding the XPS data analysis.

1. However, while performing the final checks on the manuscript we noticed issues with peak fitting the XPS data. Consider the figure 3e (Ca and Mg), the fitted curve goes beyond the acquired XPS data. Similar issues are present in most, if not all, XPS plots. Kindly recheck the peak fitting in the XPS data and revise the manuscript accordingly. Please also describe the model used to fit the data in the methods section.

Response

We thank the Editor for the detailed inspection of our data visualization. We fully agree that the cumulative fitted curve must strictly align with the acquired experimental data to ensure physical accuracy. We have rigorously re-examined the raw data and re-processed the peak fitting for Figure 3e as well as all other relevant XPS spectra included in the manuscript and Supplementary Information. We have corrected the fitting parameters—specifically optimizing the background subtraction and constraining the peak widths—to ensure that the sum of the fitted components does not exceed the intensity of the acquired raw signal. The revised figures now present a physically accurate representation of the surface chemistry. Furthermore, we have expanded the "Characterization" section in the methods to explicitly describe the peak fitting model, including the background subtraction method and the line-shape function. We appreciate your rigorous check, which has helped us improve the accuracy and quality of our data reporting.

Revised as: Line 623-625, Fig. 3e, Supplementary Fig. 4 and Supplementary Fig. 7.

Line 623-625: For XPS data analysis, the spectra were calibrated to C 1s (284.8 eV) and fitted using a Shirley background and mixed Gaussian–Lorentzian function. Doublets were constrained by fixed spin-orbit splitting and area ratios.

Fig. 3 | CEM fouling in the BMED system during urine nutrient recovery from different groups over seven batches. a SEM morphological analysis (Dominant Ca²⁺-Mg²⁺-urea co-deposits in Group A; a heterogeneous composite incorporating creatinine and uric acid in Group B; and foulants superseded by a BSA-Ca²⁺-Mg²⁺ complex in Group C.); **b** EDS quantitative detection of fouling components; **c** FTIR characterization; **d** RDF analyses of interactions between urea, Ca, Mg and CEM; **e** XPS analysis of urine fouling in Group A; **f** Scaling site variation and mechanisms: In urea-containing groups (Groups A-C), Ca²⁺ co-aggregates with urea, forming deposits predominantly on the feed side of CEM. In contrast, in the inorganic-only group (Group D), Ca²⁺ rapidly migrates through the CEM and deposits on BPM, triggered by precipitation reactions with OH⁻ generated by BPM. Mg²⁺ consistently forms scaling on the alkaline side of CEM through interaction with OH⁻ during its transport across the membrane.

Supplementary Fig. 4 | XPS characterization of C-N-O functional groups on AEM surfaces:
 Fine spectra of pristine, Group A, Group B, and Group C after BMED of different urine components.

Supplementary Fig. 7 | XPS characterization of C-N-O functional groups on CEM surfaces: Fine spectra of pristine, Group A, Group B, and Group C after BMED of different urine components.